



# Mercury accumulation in leaves of different plant types – the significance of tissue age and specific leaf area

Håkan Pleijel[1], Jenny Klingberg[2,3], Michelle Nerentorp[4], Malin C Broberg[1], Brigitte Nyirambangutse[5,6], John Munthe[4], Göran Wallin[1]

[1]University of Gothenburg, Biological and Environmental Sciences, P.O. Box 461, SE-40530 Göteborg, Sweden
[2]Gothenburg Botanical Garden, Carl Skottsbergs Gata 22A, SE-41319 Göteborg, Sweden
[3]Gothenburg Global Biodiversity Centre, Carl Skottsbergs gata 22B, 413 19 Gothenburg, Sweden
[4]IVL Swedish Environmental Research Institute Inc., P.O. Box 53021, SE-40014 Göteborg, Sweden
[5]University of Rwanda, KK 737 Street, Gikondo, Kigali, PO Box 4285, Kigali, Rwanda
[6]Global Green Growth Institute, 19F Jeongdong Building, 21-15 Jeongdon-gil, Jung-gu, Seoul 04518, Republic of Korea

Correspondence: Håkan Pleijel (hakan.pleijel@bioenv.gu.se)

**Abstract.** Mercury, Hg, is one of the most problematic metals from an environmental perspective. To assess the problems caused by Hg in the environment it is crucial to understand the processes of Hg biogeochemistry, but the exchange of Hg

between the atmosphere and vegetation is not sufficiently well characterised. We explored the mercury concentration, [Hg], in foliage from a diverse set of plant types, locations and sampling periods to study whether there is a continuous accumulation of Hg in leaves/needles over time. Measurements of [Hg] were made in deciduous and conifer trees in Gothenburg, Sweden (Botanical Garden and city area) as well as of evergreen trees in Rwanda. In addition, data for wheat from an ozone experiment conducted at Östad, Sweden, were included. Conifer data were quantitatively compared with literature data. In every case

where older foliage was directly compared with younger, [Hg] was higher in older tissue. Covering the range of current year up to four-year old needles, there was no sign of Hg saturation in conifer needles with age. Thus, over time scales of approximately one month to several years, the Hg uptake in foliage from the atmosphere always dominated over Hg evasion. Rwandan broadleaved trees had generally older leaves due to lack of seasonal abscission and higher [Hg] than Swedish broadleaved trees. The significance of atmospheric Hg uptake in plants was shown in a wheat experiment where charcoal

filtrated air lead to significantly lower leaf [Hg]. To search for general patterns, the accumulation rates of Hg in the diverse set of tree species in the Gothenburg area were related to the specific leaf area (SLA). Leaf area based [Hg] was strongly negatively and non-linearly correlated with SLA, while mass-based [Hg] had a somewhat weaker positive relationship with SLA (both relationships with $p < 0.001$). An elaborated understanding of the relationship behind [Hg] and SLA would support large-scale modelling of Hg uptake by vegetation and Hg circulation in general.





## 1 Introduction


Mercury (Hg) is widely distributed in the atmosphere and is known to deposit in ecosystems, where it can transform to highly toxic methylmercury. Important atmospheric sources of Hg are for example burning of fossil fuels, artisanal and small-scale gold mining and non-ferrous metal and cement production, along with natural sources such as volcanic activity, fires and weathering of rocks (UN Environment, 2019). Most mercury in the atmosphere is recognized as gaseous elemental mercury

(GEM, $Hg^0$), globally distributed due to its long residence time in air (6-24 months), having background concentrations of approximately 1.5 - 1.7 ng $m^{-3}$ and 1.0 – 1.3 ng $m^{-3}$ in the Northern and Southern hemisphere, respectively (Sprovieri et al., 2016).

The bi-directional exchange of $Hg^0$ between the atmosphere and terrestrial surfaces is poorly understood. Deposition of Hg, representing the downward flux, is a combination of $Hg^0$ (dry deposition) and oxidized species of Hg, including particulate

Hg (dry and wet deposition). The re-emission of Hg from terrestrial and water surfaces, the upward flux, consist primarily of gaseous $Hg^0$ (Bishop et al., 2020). Wet deposition of Hg is widely monitored by measuring Hg in precipitation. Measuring dry deposition, however, is more challenging and includes dry deposition of both $Hg^0$ and $Hg^{II}$ on foliar surfaces washed off by throughfall (Graydon et al., 2008), vegetation $Hg^0$ uptake followed by litterfall (Risch et al., 2017), and direct dry deposition to ground surfaces. In this context, the net accumulation of Hg in tree leaves and needles can provide important information

on Hg fluxes over timescales of months to years.

A pronounced seasonality in the atmospheric concentration of elemental mercury, $[Hg^0]$, has been observed over the Northern Hemisphere, well correlated with the annual variation in the carbon dioxide concentration, $[CO_2]$ (Obrist, 2007). Jiskra et al. (2018) explored this observation and pointed out that the seasonal amplitude of $[Hg^0]$, like that of $[CO_2]$, increases with latitude. The authors suggested that terrestrial vegetation acts as global $Hg^0$ pump, removing atmospheric Hg mainly during the growing

season when plant gas exchange in terms of photosynthesis, and thus $CO_2$ uptake, is large. The observed seasonality of [Hg] in the Northern hemisphere would then be explained by seasonal vegetation uptake, rather than by variation in Hg emissions or atmospheric oxidation of $Hg^0$. Jiskra et al. (2018) used a median foliar Hg concentration of 24 ng $g^{-1}$ to estimate the total deposition of Hg to vegetation over the Northern Hemisphere. This estimate of the typical foliar Hg level emanates from Grigal (2002) and does not explicitly distinguish between e.g. trees and crops, conifers and deciduous trees or foliage of different age.

It is not fully understood how $Hg^0$ is taken up by foliage. However, Laacouri et al. (2013) suggested it to be a stomatal diffusion process due to found correlation between leaf Hg content and stomatal density. Also, non-stomatal adsorption of $Hg^0$ to cuticles surfaces has been observed (Stamenkovic and Gustin, 2009), but Frescholtz et al. (2003) concluded from experimental washing of quaking aspen (*Populus tremuloides*) that leaf surface deposition was not significant in their experiment. There is instead experimental evidence that the accumulation of Hg inside leaves is dominated by stomatal uptake of $Hg^0$ from the atmosphere

(Lindberg et al., 2007). This applies to trees (e.g. Millhollen et al., 2006; Assad et al., 2016) as well as crops like wheat and corn (e.g. Niu et al., 2011; Sun et al., 2019). In addition, observations suggest a continued accumulation of Hg over time in green leaf tissue, e.g. from younger to older leaves (Bushey et al., 2008) or needles (Wyttenbach & Tobler, 1988). The detailed



character and relative importance of processes subsequent to stomatal uptake in the accumulation of Hg in leaf tissue are not completely understood (Agnan et al., 2016). Du & Fang (1983) found that in wheat leaves $Hg^0$ was converted to divalent $Hg^{2+}$

resulting from oxidation, likely promoted by the enzyme catalase.

The mobility of Hg within plants after uptake by the leaves seems to be limited. In an experimental setup using different soil and air [Hg] exposure, Niu et al. (2011) observed that leaf [Hg] was strongly correlated to air [Hg] but not to soil [Hg], while root [Hg] was linked almost entirely to soil [Hg]. This explains why [Hg] in crops has been observed to be much higher in leaves than in fruits or seeds (Li et al., 2017; Niu et al., 2011). Fleck et al. (1999) found [Hg] in wood to be much lower than

and not closely related to [Hg] in needles of *Pinus resinosa*. Although [Hg] in wood is not a direct measure of Hg transport, this observation provides an indication of limited Hg transport.

There are also observations of $Hg^0$ emissions from leaves under certain circumstances, suggesting a compensation point, i.e. an atmospheric concentration below which net Hg emission from the leaves takes place, while above the compensation point net leaf Hg accumulation prevails (Hanson et al., 1995). More recently, Yuan et al. (2019), using stable Hg isotopes and a

branch chamber system, provided direct evidence of foliar $Hg^0$ re-emission countering foliar uptake. However, empirical evidence of the development of the Hg concentration over time in leaves suggests exposure of vegetation to elevated atmospheric levels of $Hg^0$ generally result in a net accumulation in leaves/needles (Lindberg et al., 2007). To estimate the large-scale net accumulation of Hg in vegetation, it is necessary to understand if the re-emission of Hg is quantitatively important in relation to uptake over timescales of months to years and at ambient levels of atmospheric Hg.

In most studies, the leaf concentration of Hg was expressed on a mass basis, $[Hg]_M$. However, from an ecosystem perspective it would be equally, or in some cases more relevant to use leaf area-based concentrations, $[Hg]_A$. It can be calculated through dividing $[Hg]_M$ by the specific leaf area (SLA, leaf area per unit leaf mass), a leaf trait which varies between plant functional types such as broadleaved trees and conifers (Poorter et al., 2009). The significance of $[Hg]_A$ follows from the assumption that $Hg^0$ is mainly taken up through the leaf surface i.e. stomata and cuticles as explained above. The ecosystem uptake may

therefore partly be driven by its leaf area index (LAI, unit area leaves per unit area ground). However, if [Hg] in the leaves saturates with time, the ecosystem uptake might instead be limited by the total leaf mass. In both cases, duration of the uptake period (i.e. the seasonal variation in stomatal opening and leaf longevity) will have a decisive effect on Hg accumulation.

The maximum leaf area index in a closed canopy has an upper limit set by incoming light interception (Larcher, 2003). Thus, a mature broadleaved forest and a mature conifer forest may approach similar LAI. Since conifer needles are typically thicker

(have lower SLA) than leaves, leaf mass will be larger in the conifer forest. If Hg uptake is related to leaf mass rather than leaf area, i.e. depends on SLA, the Hg accumulation could be larger in conifer forests than broadleaved, thus possibly representing a stronger Hg sink. This calls for an investigation of the variation in $[Hg]_A$ among trees in combination with exposure time to compare Hg accumulation by trees of different functional types. This line of argument was investigated by Wohlgemuth et al. (2020) by comparing conifers and broadleaved trees with respect to area-based Hg concentrations. A general understanding of

the relationship behind $[Hg]_A$ and SLA would support improved descriptions of Hg uptake by vegetation applied in large-scale atmospheric modelling and assessments its biogeochemical cycling.





In this study, [Hg] in leaves from a diverse set of plant types, locations and different sampling periods were studied in order to find out if there is an essentially continuous accumulation of Hg in leaf tissue over time. Collected conifer data were complemented with literature data to further analyse the relationship between $[Hg]_M$ and needle age. Finally, the relationship between Hg accumulation and SLA was investigated.

The hypotheses were:

1. Over time-periods of months-years, there is a continued net accumulation of Hg in foliage for all studied plant species.
2. Hg accumulation in foliage has a negative relationship with SLA.
3. Hg concentrations in tropical tree leaves are higher than in broadleaved trees of the temperate zone, due to the lack of seasonal abscission of broadleaved trees in the tropics.
4. Hg accumulation in wheat is dominated by leaf uptake from air and redistribution of Hg to other plant parts, including seeds, is small.

## 2 Sampling sites and methods

### 2.1 Sampling sites

Leaf samples were collected from trees at three different sites: i) the Botanical Garden Arboretum in Gothenburg, Sweden; ii) city area of Gothenburg, Sweden (7 sampling points) and iii) Nyungwe tropical montane forest in Rwanda. The wheat experiment was conducted at a wheat crop field at Östad säteri, 35 km north-east of Gothenburg, Sweden. These sites will hereafter be denoted Arboretum, Gothenburg City, Nyungwe and Östad, respectively. The Arboretum and Gothenburg City sites will collectively be denoted Gothenburg area. See Table 1 for an overview of sites and investigated plant species.

### 2.1.1 Gothenburg area

The Arboretum and Gothenburg city sites are located in the Gothenburg area on the south-west coast of Sweden. Gothenburg is the second largest city in Sweden with approximately 540 000 inhabitants. It has a maritime temperate climate with, for the latitude, moderately cool summers and mild winters. Annual mean temperature is 7.6 °C and annual mean precipitation is 772 mm.

The Arboretum (Botanical Garden collection of planted trees from Europe, Asia and North America) enabled comparison of a range of tree species in similar growth conditions. The distance to the closest traffic route is about 800 meters. Twelve tree species representing contrasting leaf characteristics were selected for comparison: four evergreen conifers, one deciduous conifer and seven deciduous broadleaved species (Table 1). Further details on the sampling sites in the Gothenburg area are provided in the supporting information.



**Table 1. Location, elevation and climate of the study sites as well as the plant types and species sampled.**

| Site | Location | Elevation (m a.s.l.) | Climate | Vegetation | Plant types | Species |
|------|----------|----------------------|---------|------------|-------------|---------|
| Östad säteri, Sweden | 57°57'N, 12°24'E | 66 | Temperate - moderately cool summers and mild winters | Agriculture crop | Annual cereal | *Triticum aestivum* |
| Gothenburg Botanical Garden Arboretum, Sweden | 57°40'N, 11°57'E | 70 - 100 | Maritime temperate - moderately cool summers and mild winters | Planted forest stands | Conifer, evergreen | *Abies sachalinensis, Picea jezoensis, Pinus nigra, Cryptomeria japonica* |
| | | | | | Conifer, deciduous | *Larix decidua* |
| | | | | | Broadleaves, deciduous | *Betula pendula, Quercus robur, Populus tremula, Sorbus commixta, Prunus sargentii, Fagus orientalis and Juglans ailanthifolia.* |
| Gothenburg city, Sweden | 57°42'N, 11°58'E | 0 - 60 | Maritime temperate - moderately cool summers and mild winters | Urban street and park trees, 2 sites | Conifer, evergreen | *Pinus nigra* |
| | | | | Urban street and park trees, 7 sites | Broadleaved, deciduous | *Quercus palustris* |
| Nyungwe tropical montane forest, Rwanda | 2°33'S, 29°16'E | 1600 - 2950 | Highland tropical, small seasonal temperature variation, short dry season (2 month) | Natural forest | Broadleaves, evergreen | *Afrocrania volkensii, Agauria salicifolia, Carapa grandiflora, Chionanthus africanus, Cleistanthus polystachyus, Faurea saligna, Ficalhoa laurifolia, Harungana montana, Ilex mitis, Macaranga kilimandscharica, Maytenus acuminate, Ocotea kenyensis, Ocotea usambarensis, Olinia rochetiana, Polyscias fulva, Prunus africana, Psychotria mahonii, Rapanea melanophloeos, Strombosia scheffleri, Syzygium guineense* |




### 2.1.2 Nyungwe tropical montane forest

Nyungwe tropical montane rainforest is a National Park located in South-Western Rwanda, ranging from 1600 to 2950 m a.s.l. and covering an area of 1013 km$^2$. It is the largest remaining middle-elevation montane rainforest in central Africa. The forest consists of a mixture of late successional and early successional forest with patches of savannah, bamboo groves and marshes.

Annual mean temperature in sampled areas is 13.7 to 15.6°C and mean annual precipitation is 1867 mm. The seasonal variation in temperature is small but precipitation varies both spatially and seasonally, with a dry period of two months, normally from mid-June to mid-August (Nyirambangutse et al., 2017). Sampled trees were growing at elevations between ca 1950 and 2500 m a.s.l. along a 32 km-long east-to-west transect. Nyirambangutse et al. (2017) provides details of the sampling sites and characteristics of the tree species.

### 140 2.1.3 Östads säteri, south-west Sweden

Wheat data were obtained from an experiment in which field grown wheat was exposed to five different levels of ozone in open-top chambers. The treatments were: charcoal filtered air (daytime average ozone concentrations in brackets): CF (7 ppb ozone), non-filtered air: NF (20 ppb), and non-filtered air with three levels of elevated ozone: NF+ (34 ppb), NF++ (48 ppb), NF+++ (62 ppb). The site is located at Östads säteri (Table 1). Each treatment was replicated with five chambers (n=5). Ozone

exposure started on 1 July and continued until harvest by the end of August. Mid-anthesis (flowering) was reached on 8 July. Gelang et al. (2000) provides details of the experimental system.

### 2.2 Collection and preparation of samples

Three trees were sampled of each species at all sampling points in the Gothenburg area while three to 38 trees were sampled of each species in Nyungwe (combined into 2-11 composite samples, 60% of the species included > 10 trees). A pruning pole

was used at all sites to cut branches from the upper part of the tree crown. Branches with leaves/needles in the outer part of the crown were selected. In the Gothenburg area leaves from broadleaved deciduous trees including *Larix decidua* were collected 25-28 June and 17-20 September 2018, while shoots from conifers of the age classes current+1 (C+1) and current+3 (C+3) were collected on 25-28 June during the same year. Mature leaves from Nyungwe trees were collected in the period between August and December 2013. The samples were packed in polyethylene plastic bags, transported in a cool bag and stored in a

fridge until further handling.

Five to six leaves from each sampled broadleaved tree and three shoots of each age class and conifer tree were randomly selected from harvested branches for determination of SLA. A total of six or more leaf discs of a known area (18, 13, 10 or 8 mm in diameter, depending on leaf size and sampling location) were collected per sample using a puncher. The discs were oven-dried in 70°C for at least 48 hours and then weighed (laboratory balance with 0.1 mg resolution). For conifers, 20 needles

(40 from *Larix* due to small needle size) were removed using tweezers and scanned to determine total projected area using WinSEEDLE (plant image analysis scanner and software from Regent Instruments Inc, Canada; version Pro 5.1a). Needles





were dried and weighed, following the same procedure as for broadleaved species. Determination of SLA for *Cryptomeria* was complicated by the shoot/needle morphology prohibiting complete consistency with the other conifers. For this species, the needles could not be removed from the shoots without causing damage and the projected area of the shoot including needles was determined and values are therefore not completely comparable to the other species.

Sampled wheat plants were separated into leaves, straw, grain and chaff. Three shoots were sampled on each of three days (16, 18, 21 July) 1999 in the early part of the grain filling period and on three days (13, 15, 18 August) short before final harvest (starting 19 August) from each plot. The samples for each experimental plot from the three earlier dates and the three later dates, respectively, were pooled to obtain sufficient plant material for elemental analysis of all fractions (seeds, leaves, straw, chaff), the two sampling periods thus being approximately four weeks apart.

Samples for element analysis, carefully collected to avoid contamination, were put in aluminium (Gothenburg area and Östad) or paper envelopes (Nyungwe) and dried in 70°C until constant weight (at least 48 hours). Thereafter the samples were ground to a fine powder using a ball mill (model MM 301, Retsch, Haan, Germany) equipped with grinding jars and balls made of wolfram carbide, except samples from Östad that were ground with jars and balls made of stainless steel.

**2.3 Analysis of the concentration of Hg and other elements in leaves and needles**

Samples were analysed for the content of 37 elements including Hg using inductively coupled plasma mass spectrometry (Dry Vegetation ICP-MS, 37 elements after digestion in $HNO_3$ and then aqua regia (Method VG101); Bureau Veritas Mineral laboratories, Vancouver, BC, Canada). Here, only Hg data are presented. The concentrations of Hg in the samples were determined as mass-based $[Hg]_M$ and were also converted to area-based $[Hg]_A$ by dividing $[Hg]_M$ with SLA. The laboratory has implemented an analytical quality management system meeting the requirements of ISO/IEC 17025 and ISO 9001. The method detection limit (MDL) for Hg was 1 ng g$^{-1}$. Values below MDL, obtained for samples of grain, stems and chaff of wheat, are presented as 0.5MDL and were excluded from statistical analysis. All other samples were >MDL.

To test if there is an influence of drying temperature on the estimation of [Hg], a batch of *Populus tremula* leaves were collected in the Arboretum in August 2019 and three subsamples were dried at each of the following temperatures: 20, 30, 40, 50, 60, 70, 80 and 90°C and were then subject to elemental analysis as described above. As can be inferred from Fig. 1, there was no indication of an influence of drying temperature on the estimated leaf $[Hg]_M$.



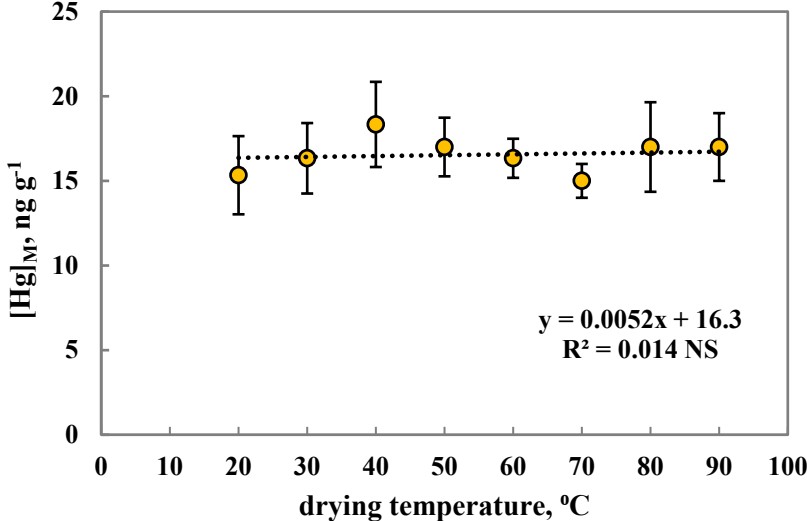

**Figure 1. Mass based concentrations of Hg, $[Hg]_M$, for one batch of *Populus tremula* leaves in relation to drying temperature. Error bars show standard deviation among the three replicates of each temperature. The linear regression was non-significant (NS).**

**2.4 Literature review for $[Hg]_M$ of conifer needles**

The scientific literature was searched for data on [Hg] in conifer needles using Web of Science. Only studies with the age classes of sampled needles explicitly specified were included. A summary of conifer needle Hg data found in literature is presented in Table 2. The specific data is presented in the file available at the data repository. Data only presented in graphs were extracted using GetData Graphic Digitizer software (version 2.26).

**Table 2. Overview of the data obtained from the literature search and our study to investigate the dependence of $[Hg]_M$ on needle age class, including literature reference, species studied, country where the sampling took place, number of sites covered by the study and needle age classes sampled. C, current year needles, C+1, 1-year old needles etc.**

| Reference | Species | Country | Sites | Needle age classes |
|---|---|---|---|---|
| Rasmussen et al. (1991) | *Picea glauca* *Abies balsamea* | Canada | 1 | C, C+1 |
| Rasmussen (1995) | *Picea glauca* *Abies balsamea* | Canada | 1 | C, C+1, C+2 |
| Fleck et al. (1999) | *Pinus resinosa* | USA | 3 | C, C+1 |
| Hutnik et al. (2014) | *Pinus nigra* | USA | 1 | C, C+1, C+2 |
| Navrátil et al. (2019) | *Picea abies* | Czech republic | 2 | C, C+1, C+2, C+3, C+4 |
| Wohlgemuth et al. (2021) | *Picea abies* *Pinus sylvestris* | Finland Germany Norway Sweden Switzerland | 10 | C, C+1 (11 data sets) C, C+1, C+2 (1 data set) C, C+1, C+2, C+3 (4 data sets) |
| This study | *Pinus nigra* *Picea jezoensis* *Abies sachalinensis* *Cryptomeria japonica* | Sweden | 3 1 1 1 | C+1, C+3 |





### 2.5 Calculation of the annual Hg uptake rate by leaves and needles in the Gothenburg area

For broadleaved trees and *Larix*, September values of $[Hg]_M$ and $[Hg]_A$ were considered to represent the Hg uptake during one growing season, since the sampling was made short before senescence and shedding of leaves/needles. To estimate the uptake of Hg by perennial needles per growing season, C+3 values of $[Hg]_M$ and $[Hg]_A$ were divided by three. C+3 needles collected by the end of June are marginally older than three years.

### 2.6 Statistical methods

Differences in $[Hg]_M$ and $[Hg]_A$ for the sampling in the Gothenburg area were investigated using a mixed designed analysis of variance (ANOVA) with IBM SPSS Statistics (version 25). Leaf/needle age was set to the within-subjects variable (repeated measures) with two levels, the two different shoot age classes for conifers and the two sampling times for deciduous trees as well as wheat. The between-subjects factor was set to either sampling location in Gothenburg City, species in the Arboretum or experimental ozone treatment of wheat. Tukey´s HD was used as post-hoc test. Linear regression was used to assess the

relationship between $[Hg]_M$ and drying temperature, between $[Hg]_M$ and needle age, using absolute concentration values as well as concentrations in relation the needle age class C+1. Non-linear regression was used to study the relationship of $[Hg]_M$ and $[Hg]_A$ with SLA. One-way ANOVA was used to investigate variation in $[Hg]_M$ among tree species in Rwanda using Hochberg's GT2 as post-hoc-test. The difference in $[Hg]_M$ between Rwandan trees and September values of broadleaved trees in the Gothenburg area was tested using the non-parametric Mann-Whitney U-test.

## 3 Results

### 3.1 Conifers in Gothenburg area

$[Hg]_M$ and $[Hg]_A$, respectively, of C+1 and C+3 needles of the different species of conifers with perennial needles, sampled in the Botanical Garden Arboretum, are presented in Fig. 2a and 2b. There was a significant (p<0.001) and consistent increase in Hg concentration for both $[Hg]_M$ and $[Hg]_A$ from C+1 to C+3 needles. For $[Hg]_M$, the variation among species was modest,

with no significant differences, but for $[Hg]_A$ there was a significant difference among species, where the post-hoc test showed that *Cryptomeria* was significantly different from *Abies*.

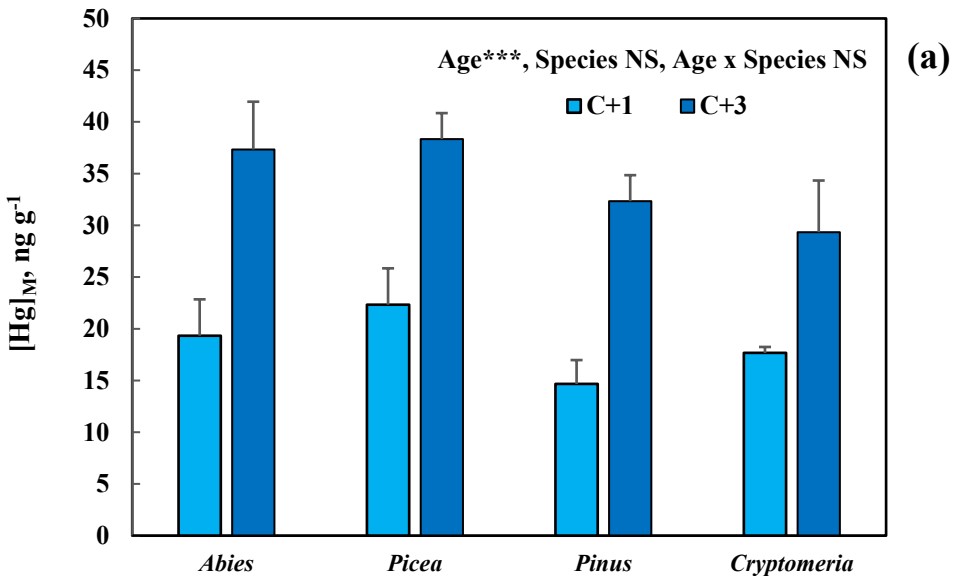

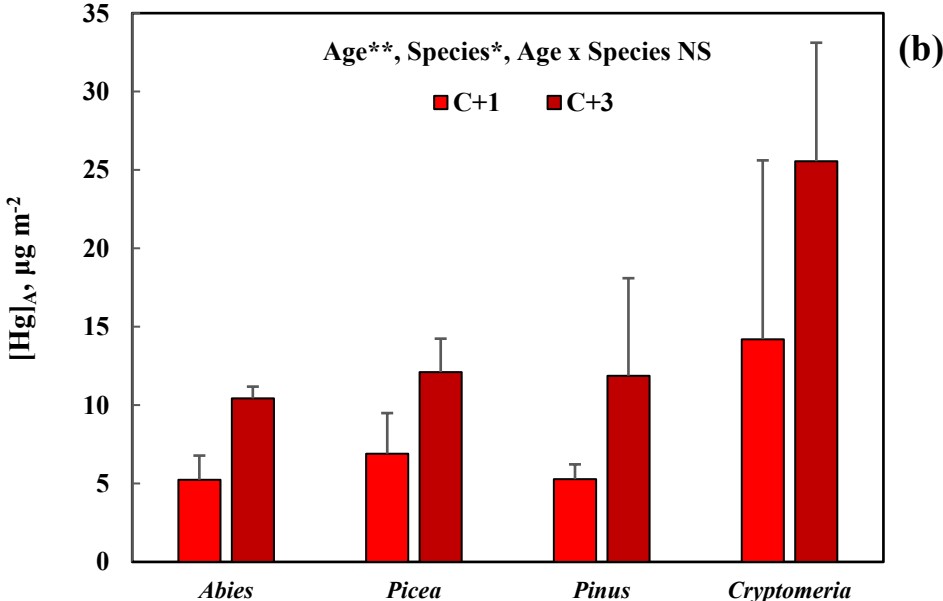

**Figure 2. Mass-based [Hg]$_M$ (a) and area-based [Hg]$_A$ (b) in 1-year old (C+1) and 3-year old (C+3) needles of conifers (*Abies sachalinensis*, *Picea jezoensis*, *Pinus nigra*, *Cryptomeria japonica*) in the Gothenburg Botanical Garden Arboretum. Error bars show standard deviation for the three trees sampled per species; \*, p<0.05; \*\*, p<0.01; \*\*\*, p<0.001; NS, non-significant**






For comparison between sites, data for $[Hg]_M$ and $[Hg]_A$ in *Pinus nigra* are here presented at three sites in the Gothenburg Area, Fig. 3a and 3b, respectively. Again, the concentration increases from C+1 to C+3 needles was consistent and significant ($p < 0.001$), while the variation between sites was small and non-significant for both $[Hg]_M$ and $[Hg]_A$.

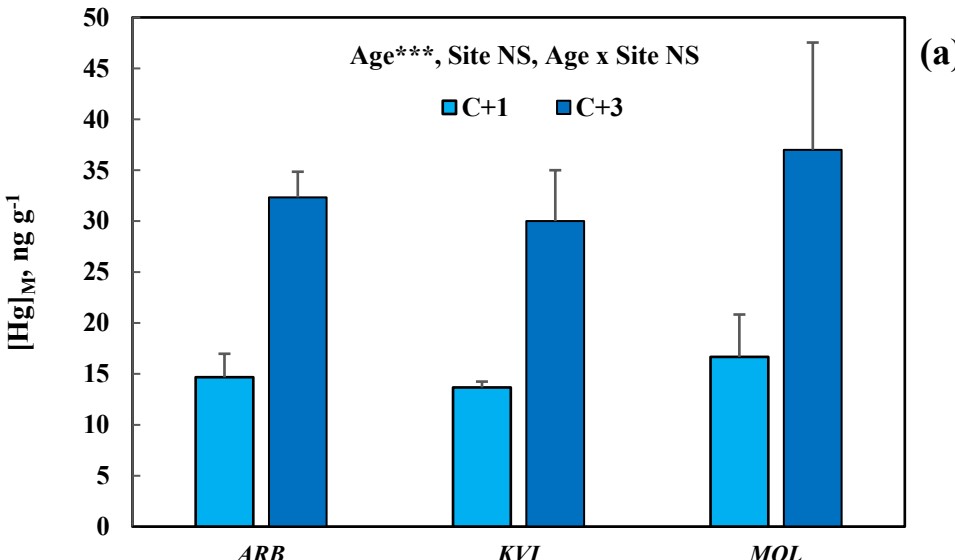


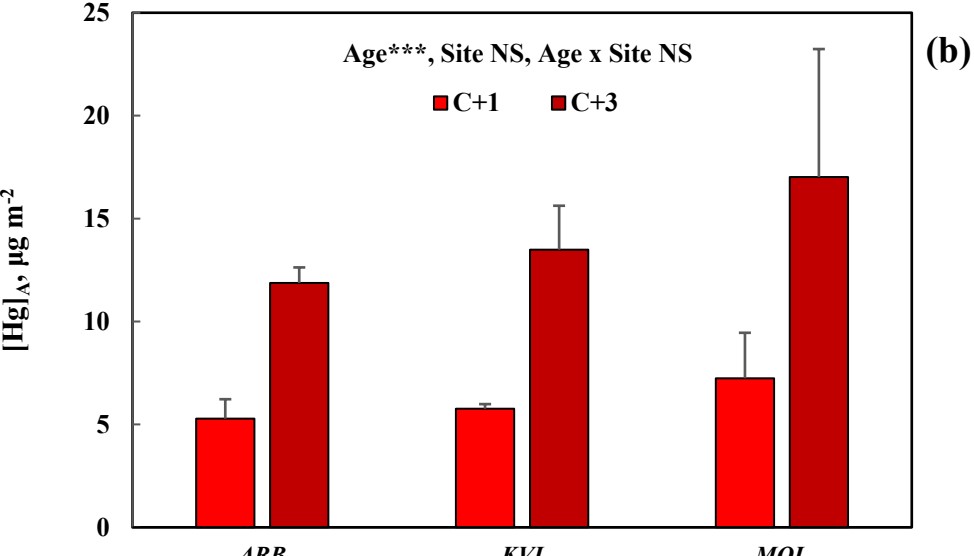

**Figure 3. Mass-based $[Hg]_M$ (a) and area-based $[Hg]_A$ (b) in 1-year old (C+1) and 3-year old (C+3) needles of *Pinus nigra* at three sites in the Gothenburg area including Arboretum. Error bars show standard deviation for the three trees sampled per species. \*\*\*, p<0.001; NS, non-significant. Site locations and characteristics are available in the supporting information.**





### 3.2 Conifer data from the literature


In Fig. 4a, the data regarding conifers from our investigation in the City of Gothenburg is combined with literature data. There was a strong and highly significant linear relationship between $[Hg]_M$ and needle age, although the data comes from different parts of Europe and North America and represent ten different conifer species of the four genera *Abies*, *Picea*, *Pinus* and *Cryptomeria*. In Fig. 4b, relative values of $[Hg]_M$ in different needle age classes is shown using the most commonly sampled

needle age class, C+1, as the reference. Values for the other age classes of a certain data set were divided by the value for C+1 for that data set to obtain a common, relative scale for all observations. Data sets without C+1 data were consequently not included. Likewise, C+4 needles were not included since only one paper reported data for this needle age class. There was no indication of a saturation of $[Hg]_M$ at higher needle age. Instead, the amount of Hg added per year is very similar from one needle age class to the next, over the range of needle age classes covered, both in Fig. 4a and 4b.

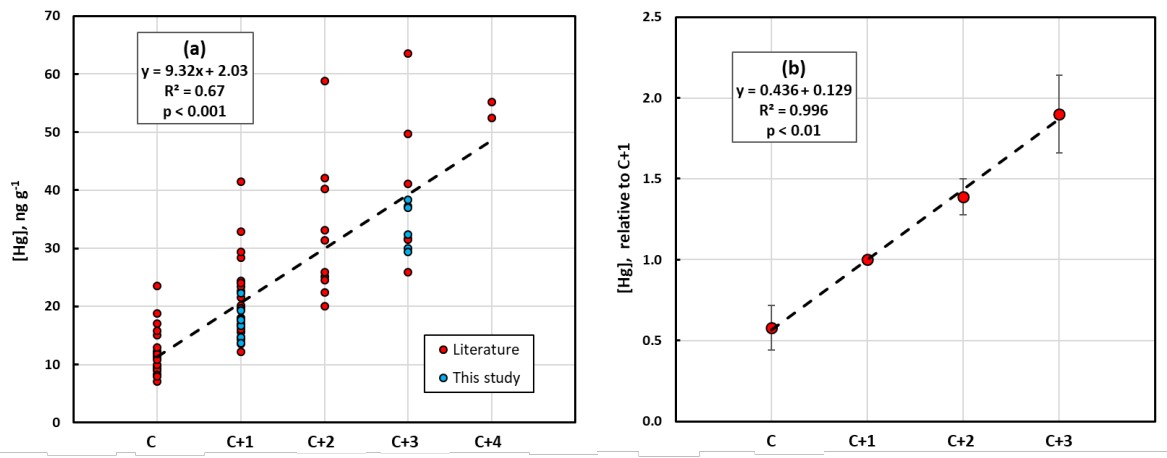


**Figure 4. (a) Absolute mass-based concentration of mercury, $[Hg]_M$, in different conifer species from the present study and data from the literature in relation to needle age (C, current year; C+1, 1-year old; etc). (b) Averages of mass-based concentration of mercury, $[Hg]_M$, from the present study and from the literature in relation to needle age as values relative to the concentration in the 1-year old (C+1) needle age class. For each dataset the $[Hg]_M$ of the different needle age classes was divided by the $[Hg]_M$ of the**
**C+1 to create a relative scale. Error bars show standard deviation.**

### 3.3 Deciduous trees in Gothenburg area

$[Hg]_M$ and $[Hg]_A$ increased consistently and similarly for all broadleaved tree species, as well as for the deciduous conifer *Larix*, investigated in the Arboretum (Fig. 5). The difference in concentrations between June and September was statistically significant ($p<0.001$). There was also a significant variation among species ($p<0.01$ for $[Hg]_M$ and $p<0.05$ for $[Hg]_A$), but no

significant interaction between leaf age and species was obtained in the statistical analysis. The post-hoc test showed that for $[Hg]_M$ *Betula* and *Larix* were significantly different from *Fagus* and in the case of $[Hg]_A$ *Prunus* differed significantly from *Larix* and *Quercus*.

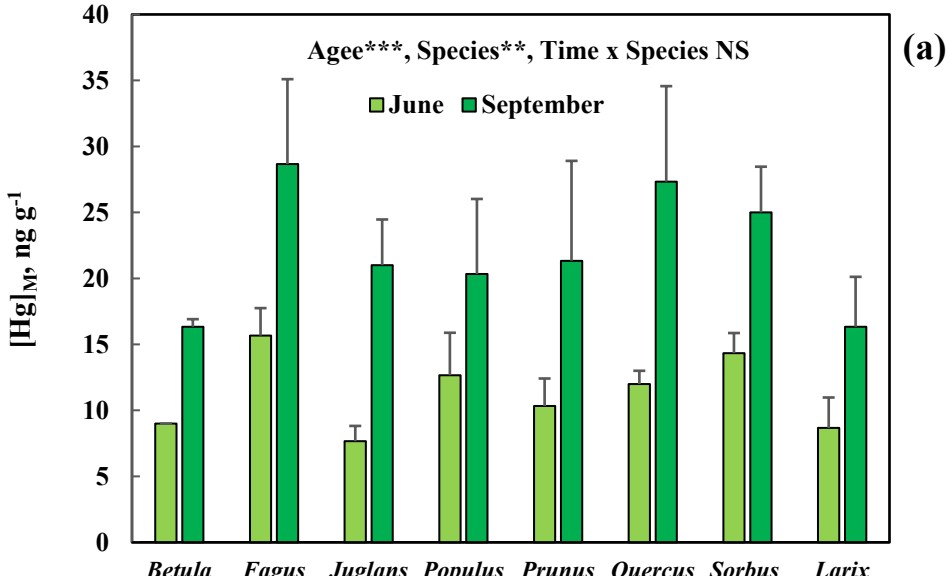

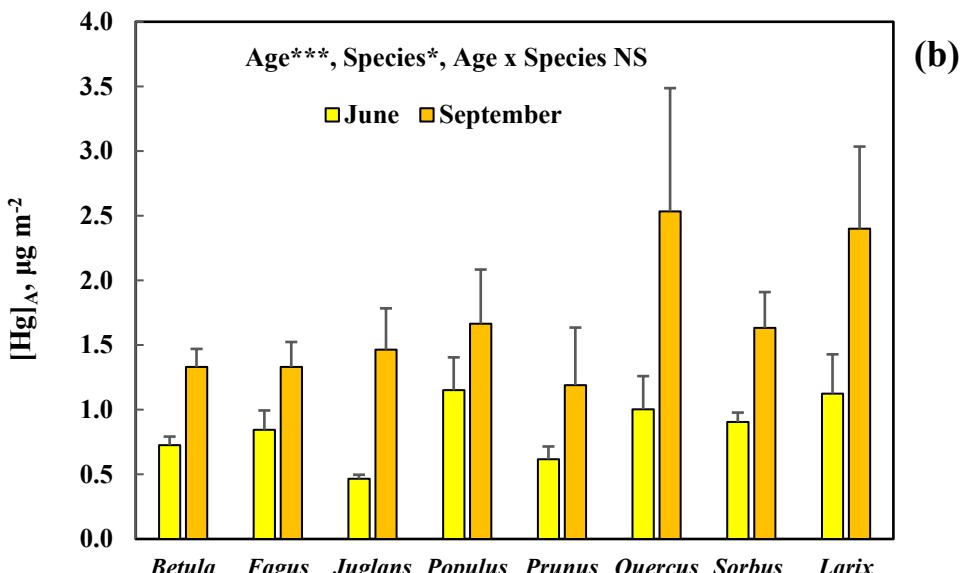

**Figure 5. Mass-based [Hg]$_M$ (a) and area-based [Hg]$_A$ (b) concentrations of mercury in leaves sampled in June and September of seven broadleaved tree species (*Betula pendula*, *Fagus orientalis*, *Juglans ailanthifolia*, *Populus tremula*, *Prunus sargentii*, *Quercus robur*, *Sorbus commixta*) and one deciduous conifer (*Larix decidua*) in the Gothenburg Botanical Garden Arboretum. Error bars show standard deviation for the three trees sampled per species; \*\*\*, p<0.001; \*\*, p<0.01; \*, p<0.05; NS, non-significant.**

[Hg]$_M$ and [Hg]$_A$ in *Quercus palustris* at seven different sites in the City of Gothenburg (Table 1) showed a consistent, highly significant (p<0.001) pattern of increased concentrations from June to September (Fig. 6). The concentration levels were





similar at all the different sites and consequently the site effect was non-significant (p>0.05). The smaller difference among sites with *Quercus palustris* compared to the difference among species in the Arboretum may represent genetic variation in Hg uptake rate among the investigated species in the Arboretum. The genetic variation is expected to be small when only one species is considered.

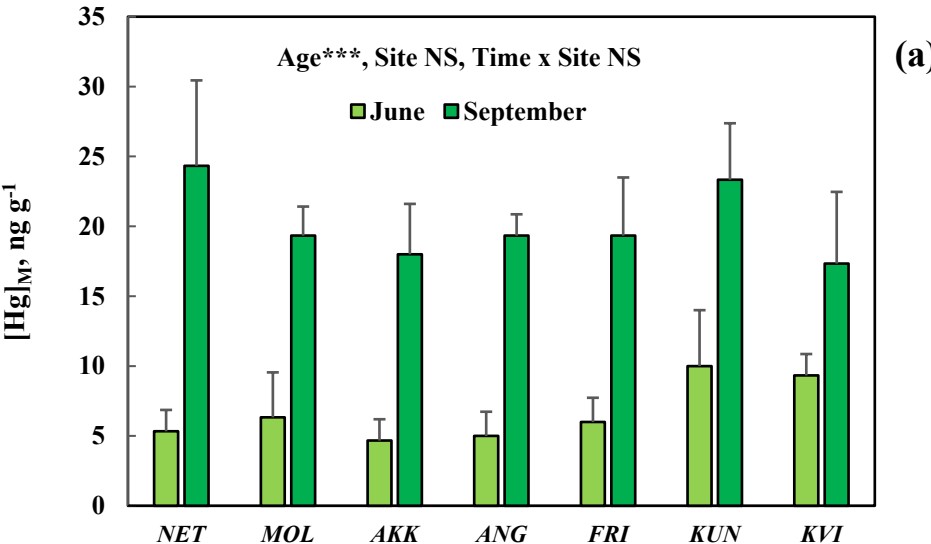


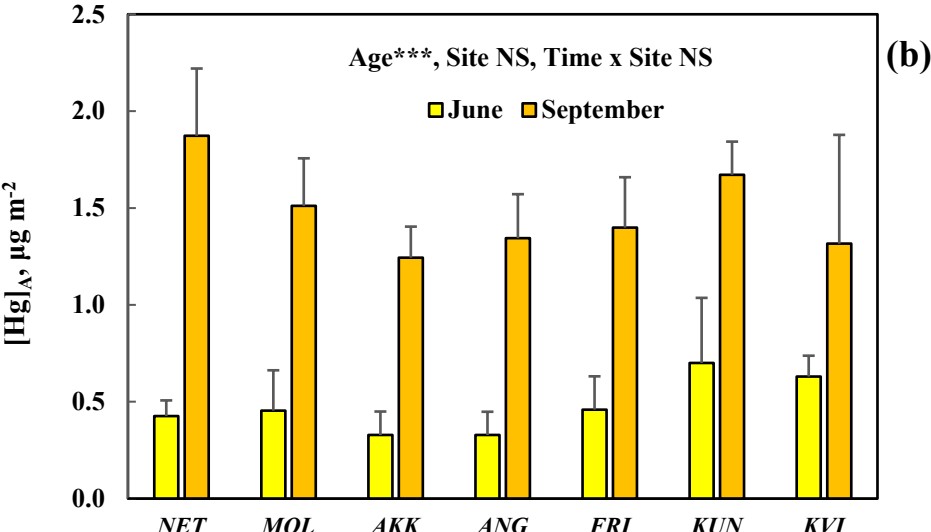

**Figure 6. Mass-based [Hg]$_M$ (a) and area-based [Hg]$_A$ (b) concentrations of mercury in leaves sampled in June and September from *Quercus palustris* trees at seven sites with different local air pollution in the City of Gothenburg. Error bars show standard deviation, between the three trees sampled per species; \*\*\*, p<0.001; NS, non-significant. Site locations and characteristics are available in the supporting information.**




### 3.4 Relationship between annual Hg accumulation and specific leaf area SLA

As explained in the Methods section, September values for broadleaved trees and *Larix* were considered to represent the Hg uptake during one growing season. For conifers with perennial needles C+3 values were divided by three to obtain a comparable growing season estimate of Hg uptake. The resulting $[Hg]_M$ and $[Hg]_A$ data for both groups of trees were plotted

vs. SLA for all observations in the Gothenburg area (Fig. 7). There was a non-linear, consistent ($R^2 = 0.87$) relationship between the annual increment in $[Hg]_A$ and specific leaf area SLA (Fig. 7a). The rate of increase in $[Hg]_A$ per growing season was larger for evergreen conifers, with low SLA, compared to broadleaved trees, having higher SLA. The annual conifer *Larix* was intermediate between these groups. For the growing season rate of increase in $[Hg]_M$, there was a weaker but statistically significant positive relationship with SLA (Fig. 7b), where the broadleaved trees consistently showed higher rates than

evergreen conifers. Again, *Larix* was intermediate. Thus, the evergreen conifers had higher per unit leaf area uptake rates of Hg strongly dependant on SLA, while broadleaved trees had higher Hg uptake rates per unit leaf mass which were more variable and not as strongly linked to SLA.

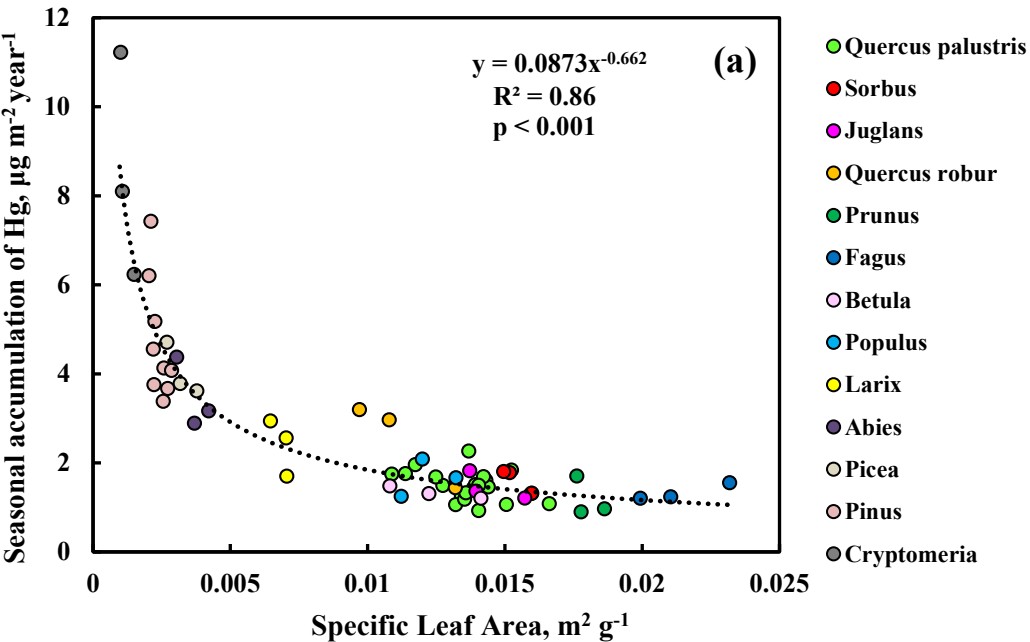





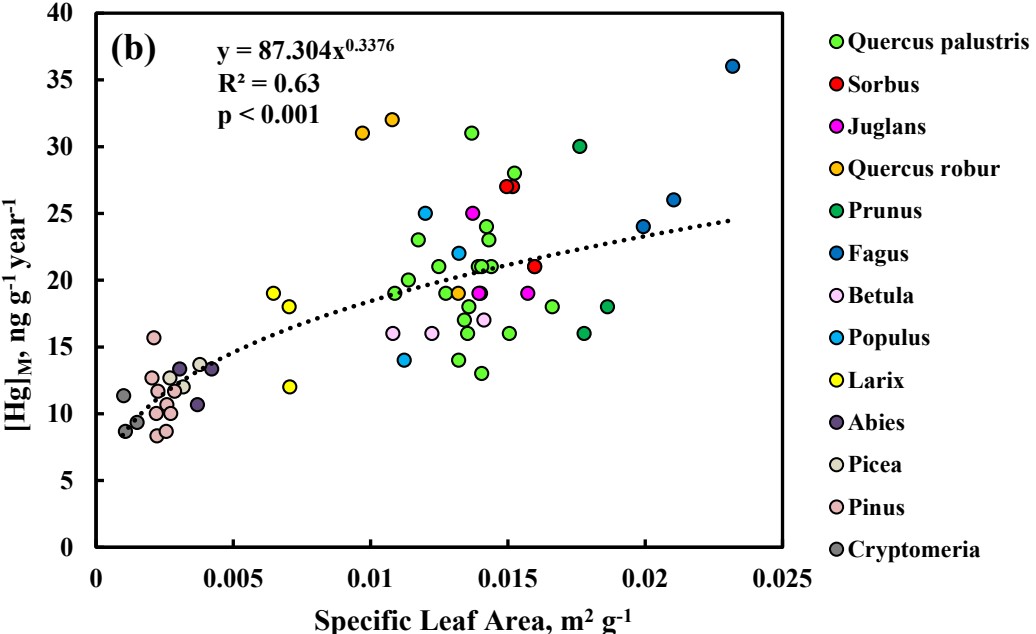

**Figure 7. Seasonal accumulation rate of Hg (a) per unit leaf area and (b) per unit dry mass in relation to specific leaf area. Each data point represents the observation for an individual tree.**

### 3.5 Broadleaved evergreen trees in Nyungwe

$[Hg]_M$ was investigated in leaves from 20 broadleaved evergreen tree species in Nyungwe (Table 1). As evident from Fig. 8, there was a substantial variation between species and the ANOVA showed that there was statistically significant variation ($p<0.01$) in $[Hg]_M$ among species, as displayed in Fig. 8. The post-hoc test revealed that there were significant differences between the species with the highest $[Hg]_M$ (*Ocotea usambarensis*) and eleven of the other species as depicted in Fig. 8. None of the other differences between species were statistically significant.

The average $[Hg]_M$ for all species of the Rwandan data set was 43.9 ng $g^{-1}$ with a standard deviation among species of 17.3 ng $g^{-1}$. This can be compared with the average $[Hg]_M$ for all broadleaved species in the Gothenburg area, 9.2 ng $g^{-1}$ and 21.5 ng $g^{-1}$ for June and September, respectively. Thus, in general leaf concentrations of broadleaved trees in Rwanda were approximately twice as high as in the Gothenburg area by the end of the growing season in September. The Mann-Whitney U-test showed that $[Hg]_M$ in leaves of Rwandan tree species were significantly higher ($p<0.01$) than leaves of broadleaved trees in the Gothenburg area from the September sampling.

The seasonal accumulation of Hg was not possible to determine for the Rwandan trees, as the leaf ages were unknown, but most of them were likely in the range of 0.5 to 2 years. However, their SLA were in a rather narrow range with values between most of the evergreen conifers and deciduous trees of the Gothenburg area, 0.006 to 0.012 m$^2$ g$^{-1}$. No significant relationship





between $[Hg]_M$ and SLA was observed ($R^2 = 0.03$, p = 0.45), possibly reflecting the much more narrow range of SLA for the Rwandan trees compared to those in the Gothenburg area.

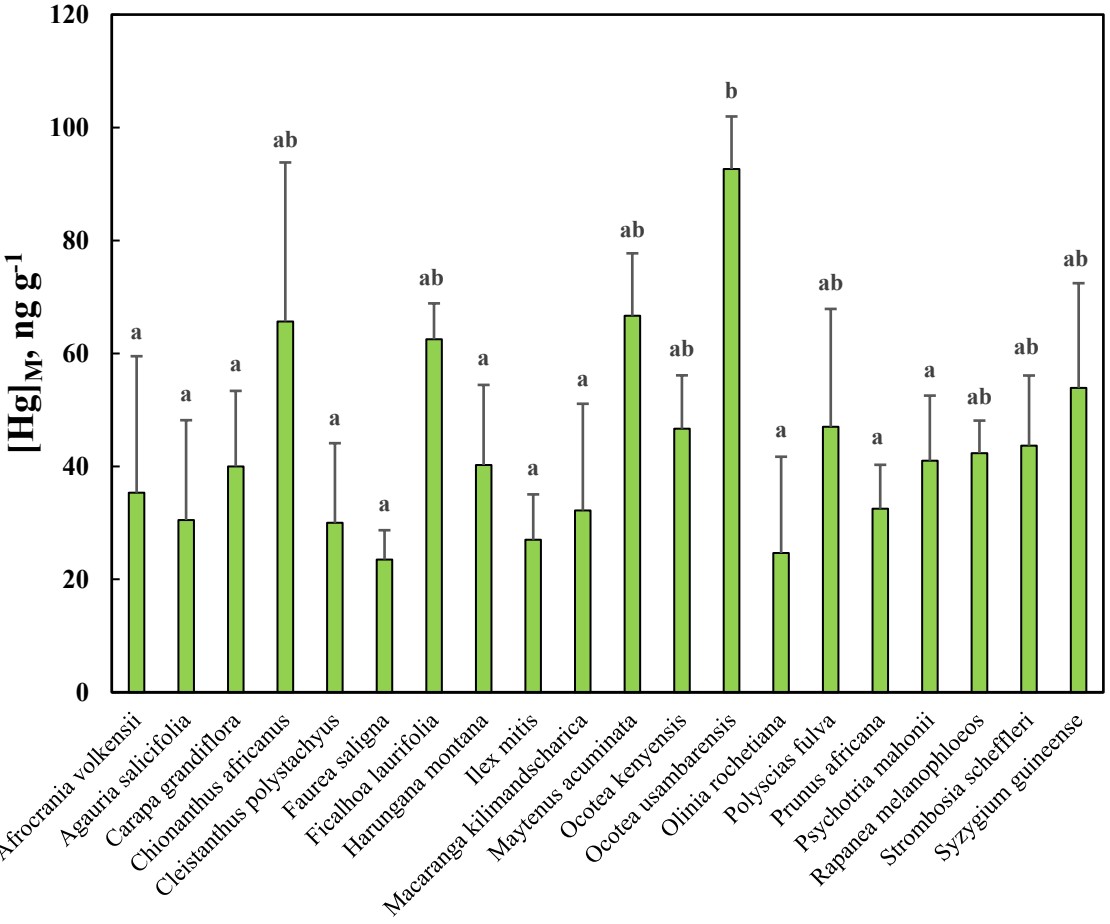


**Figure 8. Mass based concentration of mercury, $[Hg]_M$, in leaves of 20 tree species in Nyungwe tropical montane forest, Rwanda. Tree species denoted with entirely different letters are significantly different from each other according to the post-hoc test of the ANOVA.**

### 3.6 Wheat

The observations of $[Hg]_M$ in different fractions of wheat short after anthesis and at maturity in five different ozone treatments showed a number of clear characteristics (Fig. 9). $[Hg]_M$ of grain and straw was low and in several cases below the MDL of 1 ng g$^{-1}$, both after anthesis and at maturity. Grains had the lowest Hg levels. For chaff, the concentrations were somewhat higher and above MDL for all samples at maturity. In leaves, $[Hg]_M$ was above MDL in all samples and higher than in the other plant fractions. The ANOVA was performed only for leaves, the plant fraction for which all samples were above MDL for both

sampling periods. It showed a statistically significant increase in $[Hg]_M$ from anthesis to maturity (p<0.001), a significant ozone treatment effect (p=0.002) and a significant interaction time by treatment effect (p<0.001). Post-hoc test revealed that





$[Hg]_M$ of the charcoal filtered treatment, which did not increase over time, was significantly different from the other four ozone treatments among which there were no statistically significant differences. This can be explained by the removal of Hg by the charcoal filters and shows that the uptake of $Hg^0$ from the air was the main source of leaf Hg. In the other four treatments,

there was an increase in $[Hg]_M$ between the two sampling periods. For the non-filtered air treatment, representing the current ambient air situation, there was approximately a doubling of $[Hg]_M$ between the sampling after anthesis and that before harvest. The magnitude of this leaf age effect on $[Hg]_M$ declined with increasing ozone concentration, likely associated with the shortening of leaf life span from ozone, reducing the duration of the period with gas exchange of the leaves. Regressing the leaf $[Hg]_M$ at maturity vs. the ozone exposure index AOT40 (Accumulated exposure Over and ozone concentrations threshold

of 40 ppb; Fuhrer et al., 1997) resulted in a significant ($p = 0.003$, $R^2 = 0.994$) relationship ($[Hg]_M = 50.9 - 0.017*AOT40$).

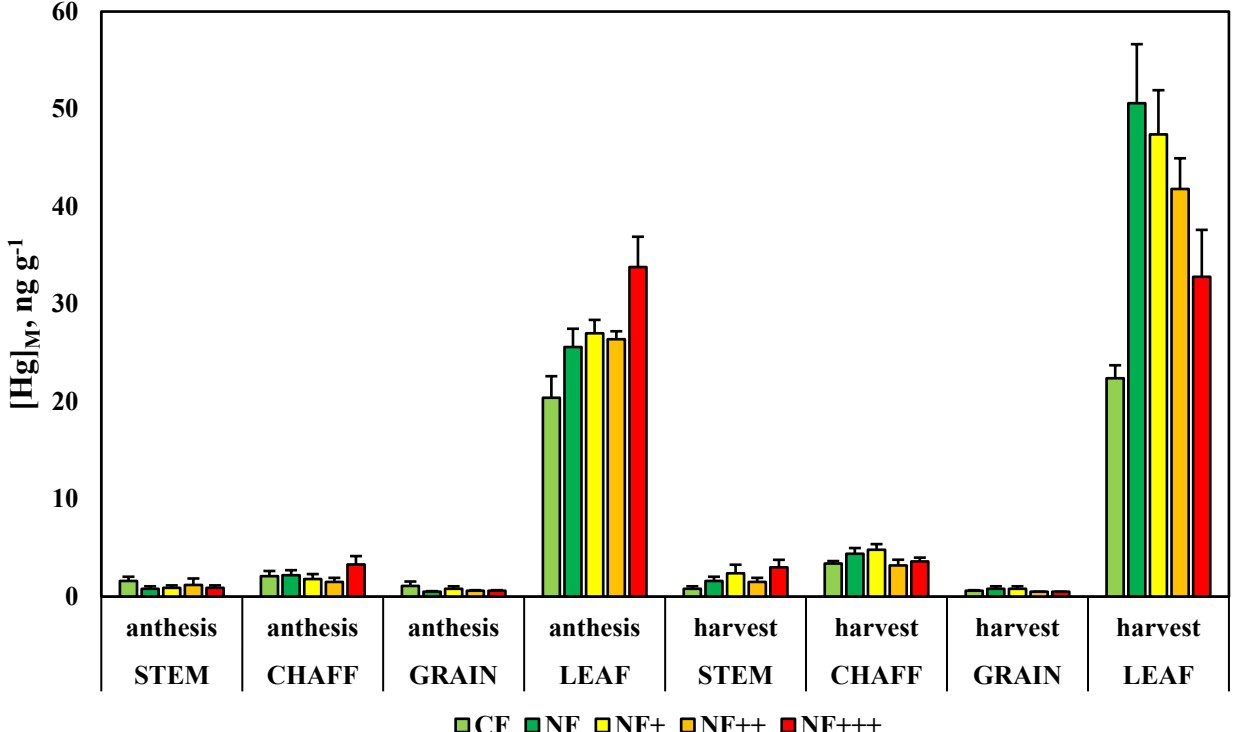

**Figure 9. Mass based concentrations of mercury, $[Hg]_M$, in four fractions of wheat after anthesis and short before harvest. Data are from an ozone experiment with field grown wheat. CF, charcoal filtered air; NF, non-filtered air, NF+, NF++ and NF+++, non-filtered air with different levels of ozone added. Error bars show standard deviation for the five replicate experimental chambers**
**per treatment.**

**Discussion**

Our data from the Gothenburg area showed, with no exception, that Hg accumulated over time in leaf and needle tissue in different tree species at varying locations and without signs of saturation with time. The literature data for $[Hg]_M$ in needles of





different age (Fig. 4a and 4b) likewise showed a monotonic pattern of increase over time, which agreed well with our data and
suggests that over the time-scales considered here, months to years, net accumulation of Hg over time is the rule. Our results
also agree with earlier observations (e.g. Fleck et al., 1999; Milhollen et al., 2006; Hutnik et al., 2014; Laacouri et al., 2013),
which together points towards a consistent and clear pattern of Hg accumulation in foliage over time, in line with our first
hypothesis.

Based on a range of literature sources, Grigal (2002) reported that most tree foliar $[Hg]_M$ concentrations are in the range 10-
50 ng g$^{-1}$ with a midpoint of 24 ng g$^{-1}$. This value was used by Obrist (2007), and later by Jiskra et al. (2018), in the assessment
of the role of vegetation in the annual dynamics of the atmospheric Hg concentration. Essentially, this range and midpoint
value are in line with our observations of all trees measured for $[Hg]_M$ by us, with an average of 21.6 ng g$^{-1}$ and only a few
values above 60 ng g$^{-1}$. However, such a gross average conceals the dynamics of Hg accumulation in vegetation, the difference
between evergreens and deciduous trees, between biomes and the distinction between $[Hg]_M$ and $[Hg]_A$ which is relevant from
an ecosystem perspective.

Litter fall from forest trees represents an important flux of Hg to the soil system (Munthe et al., 1995; St. Louis et al., 2001).
Further, it has been reported that litter fall has higher $[Hg]_M$ than what has been measured in foliage (e.g. Rea et al., 1996). As
pointed out by Grigal (2002), this may, at least partly, be attributed to the fact that leaves shed in the autumn accumulated Hg
over the full growing season, while leaves analysed for $[Hg]_M$ were often sampled earlier during the season and are likely to
have lower $[Hg]_M$. This is in line with our observations of a consistent and substantial increase in leaf $[Hg]_M$ and $[Hg]_A$ from
June to September of broadleaved trees in the Gothenburg area. A corresponding consideration also applies to conifers, on a
multi-annual time scale, for which analyses of needle $[Hg]_M$ was made for the needle age classes C+1 and C+3. Many conifers
maintain needles 4-10 years (Reich et al., 1996, 2014) and it is mostly the oldest needles, likely having higher $[Hg]_M$, that are
shed and forming litter (e.g. Nebel and Matile, 1992, Wyttenbach and Tobler, 2000, Muukkonen, 2005). We conclude that the
bias towards sampling younger leaves/needles that had less time to accumulate Hg could, if used uncritically, lead to an
underestimation of the flux of Hg by leaf/needle shedding.

On shorter time scales release or emission of Hg from vegetation to the atmosphere may occur. Short-term closed-chamber
gas exchange experiments have provided evidence of the existence of a compensation point for Hg uptake/release from leaves
(Hanson et al., 1995) and measurements of high frequency above-canopy Hg flux using micrometeorological techniques over
field crops (e.g. Sommar et al., 2016) show that fluxes from the plant-soil system to the atmosphere occur. Further, Yuan et
al. (2019) showed that part of the Hg$^0$ that has been absorbed and oxidised in the leaves becomes reduced and re-emitted.
However, as stated by Lindberg et al. (2007), and consistent with our results, empirical evidence clearly indicates that exposure
of vegetation to Hg$^0$ in air results in a net accumulation of Hg in plant tissue over time horizons of months to years.

From a biogeochemical and ecosystem perspective there seems to be several aspects to consider when assessing the
accumulation of Hg. Firstly, the age of the leaf/needle tissue is of critical importance, with an essentially monotonic
accumulation over time as stated above. Secondly, over the range of needle age classes considered, there was no indication of
a saturation of $[Hg]_M$. This conclusion is reinforced by studies using experimentally, strongly elevated Hg exposure. Niu et al.





(2011), for example, found a linear relationship between wheat leaf $[Hg]_M$ and air Hg exposure reaching leaf $[Hg]_M$ up to above
600 ng g$^{-1}$. Barghigiani et al. (1991) observed much higher concentrations in *Pinus* needles in regions of Italy with high
atmospheric Hg concentrations than those obtained in the Gothenburg region. For example, C+2 leaves of *Pinus laricio* in Pisa
reached a $[Hg]_M$ of around 140 ng g$^{-1}$ after a steady increase from year to year, again suggesting a needle sink of Hg exceeding
Hg levels attained in our measurements.

Thirdly, it should be noted that many conifer species typically have more than three or four needle age classes (Reich et al.,
1995), while sampling of needles have focused on the youngest needle age classes up to C+2, rarely including C+3 and even
more uncommonly covering C+4. This represents a bias in the understanding of conifer accumulation of Hg towards plant
tissue with lower $[Hg]_M$ and points towards the necessity of strict protocols for which needle age classes to sample in
biomonitoring, as pointed out by Bertolotti & Gialanella (2014). Older needle age classes will provide a stronger Hg
accumulation signal in biomonitoring. Also, it suggests that studies should be undertaken where sampling is made of all needle
age classes to find out if there is a saturation of $[Hg]_M$ in the oldest needles.

The typically high longevity of evergreen tropical leaves further emphasise the importance of including older leaves in
assessments of Hg fluxes in forested ecosystems. Reich et al. (1991) showed that tropical evergreen leaves having a SLA
below 0.01 m$^2$ g$^{-1}$ had a life span between 10 – 50 months. This explains why the 20 species of tropical montane trees in
Rwandan (SLA 0.006 – 0.012 m$^2$ g$^{-1}$) on average had approximately double $[Hg]_M$ compared to deciduous trees after 5-6
month in the Gothenburg area, in line with our third hypothesis. This emphasizes the significance of the duration of uptake.
Leaves of tropical and subtropical trees of well-defined and different ages should be studied with respect to $[Hg]_M$. Trees from
tropical and subtropical biomes seem not to have been investigated for Hg concentrations to any large extent in comparison to
those from boreal regions. An exception is the investigation of air-surface Hg exchange of a subtropical forest by Zhang et al.
(2019).

Our study showed that Hg accumulation over the full range of observations for trees in the Gothenburg area had a strong
relationship with SLA, especially for $[Hg]_A$ (negative relationship), but also for $[Hg]_M$ (positive relationship). Thus, the result
was somewhat more complex than our hypothesis, only stating that Hg accumulation would be negatively related to SLA.
However, it means that SLA as a leaf trait is a strong predictor of Hg accumulation and points towards the significance of not
neglecting $[Hg]_A$ since the same $[Hg]_M$ for leaves with different SLA will result in different $[Hg]_A$. This can become useful in
modelling of Hg fluxes from the atmosphere to vegetation in forest ecosystems; rather than using a single average value like
that estimated by Grigal (2002), leaf functional traits such as SLA, and leaf longevity, could be used to fine-tune estimated Hg
uptake by different forest types. Although Wohlgemuth et al. (2020) explored the significance of $[Hg]_A$, the literature has so
far almost exclusively considered $[Hg]_M$. From an ecosystem perspective $[Hg]_A$ may be equally or more relevant as closing
canopies approaches similar LAI in different forest types. For example, a LAI of 5 m$^2$ m$^{-2}$ for an evergreen and a broadleaved
forest stand will represent the same (projected) leaf area, but the leaf mass will be larger for the evergreen (especially conifers)
stand associated with the lower SLA for evergreens and likely a stronger sink for Hg.



Further studies are required to understand the differences in Hg accumulation of conifers vs. broadleaved trees in more detail. Evergreen conifers typically have a longer growing season with gas exchange than broadleaved trees in the type of climate prevailing in the Gothenburg area. This will tend to extend the duration of Hg uptake. In addition, they have a lower SLA, which represents a larger potential sink of Hg per unit needle area. However, broadleaved trees tend to have higher stomatal conductance, which will intensify the Hg flux to the leaves. For now, it can be concluded that conifer needles have a faster increment in $[Hg]_A$ per growing season than broadleaved trees but that the $[Hg]_M$ increase is faster in leaves of broadleaved trees. Crops typically have even higher stomatal conductance than broadleaved trees (Lin et al., 2015). This explains the even faster rate of accumulation of wheat leaves in ours study. During approximately one month $[Hg]_M$ increased by 25 ng g$^{-1}$ (from 26 to 51 ng g$^{-1}$, Fig. 9), while the average increase in $[Hg]_M$ of broadleaved trees in the Arboretum was 11 ng g$^{-1}$ (from 11 to 22 ng g$^{-1}$, Fig. 5) over a three-month period.

The only species for which we analysed the distribution of Hg within the plant was wheat. It showed that leaves are the dominating, essentially ultimate sink for Hg in this species, while little redistribution to other parts of the plants seemed to occur. This is in sharp contrast to many nutrients and other elements, which are more or less efficiently redistributed from leaves and stems to grains during grain filling (Marschner, 2012). In particular, grain concentrations of Hg were very low in our study. This supports the fourth hypothesis and is in line with conclusions that leaf crops tend to have comparatively high Hg levels (De Temmerman et al., 2009) and the observation by Li et al. (2017) that $[Hg]_M$ in tomato plants were considerably higher in leaves than in fruits, stems and roots. In addition, Niu et al. (2011) reported from experiments in the Chinese Hebei Province that seed $[Hg]_M$ in wheat and maize were very low, while leaf concentrations in wheat near harvest were ~60 ng g$^{-1}$, close to our observation for wheat leaves in the non-filtered air treatment short before harvest. In the Chinese Shandong Province, Sommar et al. (2016) found $[Hg]_M$ in wheat leaves up to 120 ng g$^{-1}$. In the open-top chamber field experiment with different exposures of air and soil Hg, Niu et al. (2011) also found clear evidence that $[Hg]_M$ in foliage was correlated with air concentration, but insignificantly correlated with soil concentrations. Root concentrations, on the other hand, correlated with soil [Hg] but not with air [Hg]. The significant reduction of leaf $[Hg]_M$ by charcoal filtration, known to remove Hg$^0$, in our wheat study adds further evidence of the atmosphere as the main source of leaf Hg. An additional observation in our wheat experiment was that leaf $[Hg]_M$ near harvest declined with increasing ozone exposure. This can be explained by the shortening of the leaf life span, i.e. of the duration of leaf physiological activity and gas exchange, and thus Hg uptake, by the senescence promoting air pollutant ozone (Gelang et al., 2000).

Bishop et al (1998) investigated the xylem sap as a pathway for Hg to the canopy in *Picea abies* and *Pinus sylvestris*, with the aim to understand the contribution of this process to litter fall Hg. The authors found that only 11% of total Hg may originate from this pathway, again pointing towards shoot uptake of atmospheric Hg being the main source. Fleck et al. (1999) found, for *Pinus resinosa*, that wood concentrations were considerably lower in $[Hg]_M$ than in needles and more so in relation to C+1 needles than C needles. The conclusion from our data and the cited literature suggest that Hg has a low mobility within different types of plants.



We did not study the stomatal control of Hg uptake by leaves. However, several lines of evidence support the view that Hg in
leaves is mainly a result of stomatal uptake of atmospheric gaseous elemental mercury (Ericksen et al., 2003) and has a
correlation with stomatal density of the leaves (Laacouri et al., 2013). These authors also showed that in the investigated
deciduous species the dominating leaf sink for Hg was tissues of the leaf interior (>90%) with minor amounts on the surface
and in the cuticle. Browne and Fang (1978), using a whole-plant chamber and $^{203}$Hg-labelled mercury, found strong evidence
that all Hg vapour uptake in wheat was confined to the leaves (basically nothing in stems and roots) and that the uptake was,
like stomatal conductance, dependent on illumination but unaffected by temperature. As already mentioned, in our wheat
experiment charcoal filtration of the air during the grain filling period essentially prevented net uptake of Hg by the leaves,
pointing towards uptake from the atmosphere being the dominating Hg source. Further, Du & Fang (1982) found stomatal
control of Hg uptake in seven gramineous species and that the uptake was larger in species with the C3 compared to C4
photosynthetic pathway, consistent with the typically lower stomatal conductance of C4 plants. In line with this, Sommar et
al. (2016) observed larger fluxes of Hg$^0$ to wheat (a C3 plant) than corn (a C4 plant) fields in North China. As a consequence,
these authors reported >3 times higher [Hg]$_M$ in wheat leaves compared to corn leaves.

Our test of the influence of the drying temperature of the leaf samples did not indicate any significant temperature effects on
[Hg]$_M$. This means that higher drying temperatures in pre-treatment of samples for mercury analysis does not lead to an
underestimation of [Hg]$_M$ due to loss of volatile Hg components. This is in line with the evidence by Lodenius et al. (2003)
studying Hg accumulation in the moss *Sphagnum girgensohnii* and the grass *Lolium perenne*, as well as Wohlgemuth et al.
(2020) working with forest trees.

Our results, showing a consistent and significant accumulation of Hg in leaf/needle biomass supports the view of Jiskra et al.
(2018) that vegetation is an important Hg sink, which can explain the seasonality of background atmospheric [Hg$^0$]
concentrations in the Northern Hemisphere. This has implications for large-scale modelling of the biogeochemical Hg cycle
and for future research investigating Hg concentrations in plants. In this type of modelling, the role of the variation in SLA
among tree species and within canopies, as well as its relation to [Hg]$_M$ and [Hg]$_A$ for Hg accumulation should be considered.
In their estimation of dry deposition of atmospheric Hg to various land cover types surrounding 24 cites in North America,
Zhang et al. (2016) concluded that dry deposition of Hg to vegetated surfaces is dominated by leaf/needle uptake of Hg$^0$ (in
contrast to what has earlier been assumed) in rural environments. This finding, like our results, emphasises the importance of
the research into Hg accumulation in leaf/needle tissues.

**Conclusions**

The most important conclusions from this study are:

- Over time scales of months to years, there was always a net accumulation of Hg in leaves/needles of trees with no
  indication of a saturation in needles up to 4 year old.





•   Leaf/needle age is critical for Hg concentration, which is important for sampling strategies and in the use of plant material as a biomonitor/bioaccumulator for Hg.

    •   A bias towards sampling younger needle age classes of conifers in many earlier studies can lead to underestimation of the flux of Hg to vegetation and to soil through leaf litter fall.

    •   Tropical trees had on average higher leaf $[Hg]_M$ than leaves of temperate trees. The likely reason is the longer leaf

475          life span of tropical trees.

    •   Specific leaf area (SLA) is a leaf trait, which represents a strong predictor of unit area accumulation of Hg over the wide range of leaves/needle characteristics covered by our study of trees in the Gothenburg area. There was also a weaker, but significant positive relationship between leaf/needle mass based uptake of Hg and SLA.

    •   For certain aspects of ecosystem-level uptake, leaf area based $[Hg]_A$ is relevant to assess the tree canopy accumulation

480          capacity of Hg, typically being substantially higher in conifers than in deciduous trees.

    •   In wheat, leaves are the main sink for atmospheric Hg uptake with little transport to other parts of the plant.

### Author contributions

H.P. designed the study and performed most of the data analysis in dialogue with all co-authors. Field sampling and sample preparation was made by H.P. (wheat, trees in the Gothenburg area), J.K. (trees in the Gothenburg area), G.W. and B.N.

(Rwanda). H.P. wrote the manuscript with contributions from J.K., G.W., M.B. and M.N. All co-authors participated in discussions of the manuscript and provided comments.

### Financial support

The samples of trees in the Gothenburg area were collected within a research project studying accumulation of polycyclic aromatic hydrocarbons in urban trees funded by the research council Formas (grant: 2017-00696). ICP-MS analysis of the
leaf/needle samples was supported by the Strategic Research Area BECC (Biodiversity and Ecosystem Services in a Changing Climate, www.becc.lu.se). Analysis of crop samples was funded by the foundation Helge Ax:son Johnsons stiftelse.

### Data availability and supporting information

Original data for Hg concentrations in plant material as well as SLA data and details concerning the sampling sites in the Gothenburg area are accessible at Dryad as: doi:10.5061/dryad.7wm37pvsq

**Competing interests**

The authors declare that they have no conflict of interest.



**Acknowledgement**

We are grateful to the Gothenburg Botanical Garden and the Rwanda Development Board (RDB) for the authorization to conduct research and collect specimens in the Gothenburg Arboretum and Nyungwe National Park, respectively. We are also
grateful to the Östad Foundation for making the wheat field available to our research. Many thanks are also due to Innocent Rusizana, Pierre Niyontegereje and Etienne Zibera for great assistance in the fieldwork in the Nyungwe National Park.

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
