# Peer review of "Mercury accumulation in leaves of different plant types – the significance of tissue age and specific leaf area"

_Biogeosciences, 2021_

## Author Comment (AC2)

Suggestion for new version of Figure 7:

[Figure]

Comparison of Figure 7 a and b with separate function for conifers and broadleaved trees compared with the combined function (not intended to be included in the manuscript):

[Figure]

Relationship of [Ca]$_A$ with SLA for individual trees (as Figure 7a in our original manuscript but for Ca) and for the different species (new version of Figure 7a) – these figures are for comparison with another element and not intended to be included in the manuscript.

---

## Author Response (AR1)

*We wish to thank both reviewers for their valuable comments and suggestions and for contributing with an interesting and important discussion. Please find below our response to the reviewers, point by point.*

**Response to Anonymous Reviewer 1**

In general, the authors seem less up to date with the research front and the choice of references may seem somewhat dated. This is especially true of the obvious achievements made with stable isotope analyzes and paradigm shifts that affect the view of the mercury cycle in terrestrial ecosystems.

**Response:** *We have now included the references suggested by the reviewer. See below for more details of how the references have been incorporated in the revised manuscript.*

L69: I recommend e.g. Yang et al. PLOS ONE 2018 instead of Fleck et al. (1999) for wood analysis.

**Response:** *We agree that the recent reference is more relevant and have replaced Fleck et al (1999) by Yang et al (2018).*

L74: Yuan et al. says that re-emission is only partially counteracting uptake. There is no contradiction between Yuan et al. and Lindberg et al. (the latter is an old reference that hardly belongs here. Perhaps Bishop et al. and / or Sommar et al. both STOTEN 2020 should be referenced here) and it is difficult to understand that "however" is used in the following sense. Revise L74 - L77.

**Response:** *We agree that there is no such contradiction. The reference to Lindberg et al has been removed here, together with some associated text. Bishop et al 2020 and Sommar et al 2020 have been included. "However" has been removed and the text has been changed to "There are several observations of $Hg^0$ emissions from leaves (Bishop et al., 2020; Sommar et al., 2020). Yuan et al. (2019), using stable Hg isotopes and a branch chamber system, provided direct evidence of foliar $Hg^0$ re-emission partly counteracting foliar uptake. Empirical evidence of the development of the Hg concentration over time in leaves suggests exposure of vegetation to elevated atmospheric levels of $Hg^0$ generally result in a net accumulation in leaves/needles, which is not in conflict with the dynamical bidirectional fluxes found using high time resolution of isotope techniques (e.g. Demers et al., 2013)."*

L77: The sentence is grossly misleading. It points out the need to quantitatively examine re-emissions without mentioning that the study totally ignores doing so.

**Response:** *This sentence has been removed.*

L255: "... significant... analysis.". Can this sentence be rewritten to be easier to understand?

**Response:** *We removed the last part of the sentence: ", but no significant interaction between leaf age and species was obtained in the statistical analysis". The piece of text does not add any important information and may be confusing.*

Fig. 5a Spelling "agee".

**Response:** *Thanks, corrected.*

L337 and on: The discussion is very long without the reader being informed that there are several studies that clearly report the global distribution of atmospheric Hg uptake into vegetation and that especially the subtropics / tropics are important (eg Wang et al. 2016 EST and others). Instead, the discussion tends to delve into individual studies with unnecessary verbosity instead of measurement data being inserted and discussed in the context of the current state of knowledge (for example reported in Obrist et al. 2018 Ambio). The manuscript benefits from a sharp revision of the discussion.

**Response:** *We agree that the Discussion is long and can be shortened. In response to the detailed comments by reviewer Lena Wohlgemuth the text of the discussion has both been revised and shortened. We have also shortened the Discussion further in several places and made use of the information in Wang et al (2016) for the comparison with data from Rwanda and Sweden in our study as well as of Obrist et al (2018) for defining the context at the start of the Discussion. Some of the comments by reviewer Lena Wohlgemuth, however, also resulted in addition of text, but the Discussion as a whole, has been substantially shortened.*

L365: Here the authors' opinion from L74- is repeated. It must be said that the manuscript's lack of air analyzes is not meritorious. In what appears to be an attempt to reverse the perspective that empirical data are not compatible with extensive bidirectional exchange, a number of studies are discredited here that elegantly use stable isotope analyzes (e.g. Demers GBC 2013, Zheng GBC 2016 and Yuan EST 2019) to clearly demonstrate the importance of re-emissions from foliage. It can be repeated that the isotope studies in no way contradict continuous net uptake of mercury over the long term, on the contrary, the actual (gross) uptake is greater than what bulk analyzes of leaf samples (this study) can show.

**Response:** *We do not believe that bidirectional fluxes are inconsistent with net accumulation over time, but our phrasing was unfortunate. The text has been changed to: "Further, Yuan et al. (2019) showed that part of the $Hg^0$ that has been absorbed and oxidised in the leaves becomes reduced and re-emitted. This process is in no way inconsistent with our results that there is a net accumulation of Hg in plant tissue over time horizons of months to years. Isotope studies that have been undertaken (e.g. Demers et al., 2013, Yuan et al., 2019) show that the gross uptake of Hg by leaves is larger than what bulk analysis of leaf samples (net uptake), such as in our study, show."*

**Response to reviewer Lena Wohlgemuth**

*We appreciate the comments and suggestions from the careful reading of our manuscript. We believe that many of them improves our manuscript. However, we do not agree on all points.*

However, the authors draw far-reaching conclusions on the relation of foliar Hg uptake and specific leaf area (SLA), which cannot be easily generalized based on the regression model

and their data. I approve publication of this manuscript, if the authors adequately address and critically discuss the following issues in their data analysis:

**Response:** *As will be apparent below we will modify a number of statements about the relationships with SLA. We will retain but substantially modify the figures showing the relationships between $[Hg]_A$ and $[Hg]_M$ with SLA.*

1. In Sect. 3.4, Fig. 7b the authors portray regression results of $[Hg]_M$ to SLA values. The regression fit improves according to Fig. 7a, where ($[Hg]_M/SLA$) is regressed to SLA. However, I am concerned, that regressing a ratio to the denominator of this ratio might mathematically result in the nonlinear regression and the improvement of fit ($R^2$), that the authors display in Figure 7a, without there existing an actual relation of this kind. The authors should consider, that the chosen nonlinear regression model of y/x ~ x (Fig. 7b) is sensitive to small needle SLA values. SLA data for Crytomeria has to be excluded from the analysis, since needle area values of Cryptomeria were not measured in a comparable way to the other coniferous species (Line 165) and this might have an impact on the regression of Fig. 7. Furthermore, the data displayed in Fig. 7b looks heteroscedastic to me and p values might not be valid. The authors should also evaluate and show regression of $[Hg]_M$ and SLA separately for needles (excluding Crytomeria) and leaves before concluding that needle $[Hg]_A$ strongly depends on SLA (Line 286), since this is not obvious from a visual inspection of the data (Fig. 7b).

**Response:** *A) We have removed the Cryptomeria data from Figure 7 and Figure 2b, since the SLA determination was difficult for this species, as stated in the manuscript. The exclusion will be explained in the Materials and methods section. We have retained Cryptomeria in Figure 2a, which does not depend on the determination of SLA. B) To make the results clearer in Figure 7, we have calculated $[Hg]_A$ and $[Hg]_M$ as averages for the different species (excluding Cryptomeria); this also removes much of the heteroscedasticity and we avoid the bias from a larger number of data points for Quercus palustris and Pinus nigra compared to the other species. These figures replace the earlier Fig. 7A and B. They contain error bars showing the standard deviation both for SLA and $[Hg]_M/[Hg]_A$:*

[Figure]

*C) The figures below shows the common relationship for all species and those for conifers and broadleaved trees separately (in all cases logarithmic functions). For both Figure 7a and 7b there is a strong association with the common regression line, suggesting that it is justified to combine the data for conifers and broadleaved. We do not intend to include these figures in the manuscript.*

[Figure]

*D) It is correct that SLA is used to calculate $[Hg]_A$ (Figure 7a). It is however not necessary that a strong relationship in this type of plot results because of this. We have analysed the needles/leaves with respect to a range of elements. We exemplify this below with the macronutrient Ca, which has a substantial variation among species and site dependence. The figure to the left corresponds to Figure 7a (but for Ca) for individual trees (as in the original manuscript), to the right the version for individual species suggested for the revised version of the manuscript. These figures are not intended for inclusion in the manuscript.*

[Figure]

*We believe that, for our data set, Figures 7a and b show that SLA is of importance for explaining Hg accumulation, at least in our data set, while SLA is less or much less important for most other elements. If such a relationship exists for Hg – further investigations should be undertaken to establish this – it would be useful in modelling. As mentioned already, we will however, modify our statements about the relationships with SLA.*

2. The authors state that "SLA as a leaf trait is a strong predictor of Hg accumulation" (Line 397) and that coniferous needles take up more Hg via their surface area than broad leaves (Lines 281/282; 285/286; 410/411). While this might be true from the data presented here, I think this statement should not be generalized easily without further discussion. The authors explain in the introduction that foliar Hg accumulation is dominated by stomatal uptake during leaf diffusive gas exchange. However, stomatal conductance for leaf gas exchange is higher for broad leaves than for coniferous needles over the growing season (see abundant literature, e.g. Lin et al. 2015), which should result in higher Hg uptake via surface area of leaves compared to needles. The authors should discuss this contradiction. Furthermore, the authors should consider, that the comparison of needle versus leaf $[Hg]_A$ is sensitive to the ratio of $SLA_{leaves}/SLA_{needles}$ in association with the ratio $[Hg]_{M,needles}/[Hg]_{M,leaves}$. From Figure 7, I estimate (please ensure accessibility to raw data) a

medium $SLA_{needle}$ value of ~ 0.003 $m^2$ $g^{-1}$, which is low in comparison to literature values (see e.g. Poorter et al. 2009 or Goude et al. 2019), thus $SLA_{leaves}/SLA_{needles}$ of this study is relatively high (approximately 0.013/0.003 = 4.3). The ratio $[Hg]_{M,needles}/[Hg]_{M,leaves}$ (derived from Fig. 7) equals approximately 0.6 and roughly agrees with published literature (see e.g. Zhou et al. 2021) or is higher than values from literature (see e.g. Wohlgemuth et al. 2020). Consequently, the ratio of $[Hg]_{A,needles}$ to $[Hg]_{A,leaves}$ is roughly 4.3 x 0.6 = 2.6, meaning that in this study Hg area uptake by needles exceeds Hg area uptake by leaves by a factor of 2.6 on average. However, different and equally realistic ratios (e.g. $SLA_{leaves}/SLA_{needles}$ = 125/43 = 2.9; $[Hg]_{M,needles}/[Hg]_{M,leaves}$ = 0.4) would result in comparable uptake $[Hg]_{A,needles}$ to $[Hg]_{A,leaves}$ (e.g. $[Hg]_{A,needles}/[Hg]_{A,leaves}$ = 2.9 x 0.4 = 1.2). Therefore, please frame generalizations on the efficiency of needle vs. leaf Hg surface uptake more carefully and clearly mention caveats of the data analysis.

**Response:** *A) We agree that this conclusion is too generalized and have changed the statement on line 397 to: "… SLA as a leaf trait might be a useful predictor of Hg accumulation, which should be investigated further, ". As will be apparent below, we have also moderated the phrasing around the SLA relationships in several other places. In addition, we have added to the Discussion the following piece of text: "However, our study of the relationships of Hg accumulation in leaves and needles with SLA are based on data from one specific region with a limited number of species. Further investigations in other biogeographic areas and with further species should be made to clarify the extent to which these relationships can be generalized."*

*B) Much of the argumentation in the above comment is based on the unjustified assumption that tree species, with respect to SLA, generally can be combined in two categories: "leaves" and "needles" without consideration of which species are actually compared. It is evident from the literature that the variation within these groups is large. Even our more limited data exemplifies this: Larix, as well as the tropical trees (not included in Figure 7), have SLA values in between the broad categories of evergreen conifers and deciduous temperate broadleaved trees. Since SLA is a continuous variable, it is better to use a functional relationship between e.g. Hg accumulation and SLA instead of using broad categories for which SLA depends strongly on which species are included. Because of the large variation among species in SLA within categories, to compare our data with the literature you have to do this species by species. Pinus nigra (included in our study), for example, has a lower SLA than several other evergreen gymnosperms. In Poorter et al 2009 the 10 to 90 percentile for 70 evergreen gymnosperm species ranged between approximately 0.1 and 0.002 $m^2$ $g^{-1}$. The average SLA for evergreen conifers in our study of 0.003 $m^2$ $g^{-1}$ is the 75% percentile for evergreen gymnosperms in Poorter et al (2009).*

3. In line with 2.), please include QA of measurements of projected needle areas (Lines 159 - 165) and exclude area values for Cryptomeria as they are not comparable to area values of the other species (Line 165). Please discuss the representativeness of September values of leaf $[Hg]_A$ uptake for the whole growing season. SLA increases before leaf abscission towards the end of the growing season (see Reich et al. 1991 and Epron et al. 1996), and this effect might start as early as September at the latitude of this study. The unusual hot and dry summer of 2018 might have had an effect on leaf Hg concentrations. $[Hg]_A$ of needles C + 3 might not represent needle $[Hg]_A$ uptake over one growing season due to a

decrease of SLA with needle age (see e.g. Xiao et al. 2006). Furthermore, C + 3 needles took up Hg(0) over the course of four growing seasons, therefore concentrations of C + 3 needles should be divided by four and not three.

**Response:** *A) Our lab has been doing this kind of analysis since the late 1980s and we have described the methods used in quite some detail in the Materials and Methods section. We have cross-checked the methods we are using to determine leaf and needle areas using punchers and different image analyzers and the agreement between methods are normally within a few percent range. We are therefore confident that we can do this accurately. B) As explained above, we will remove Cryptomera data everywhere where it is associated with SLA. C) We measured the chlorophyll content of the leaves both in June and September. They showed small differences between the two sampling times (Average±SD for all broadleaved species were $0.30\pm0.036$ g m$^{-2}$ in June and $0.31\pm0.049$ g m$^{-2}$ in September). This suggests that autumn senescence had not started at the second sampling (which was also the intention of the sampling strategy). D) It is true that SLA may change over the life span of a leaf or needle. The SLA values for September were determined in direct association with the sampling. Thus, the SLA values are representative for time short before leaf shedding, which is the way it ought to be when we use leaf Hg data for that time. E) It is correct that summer of 2018 was unusually warm, also in the Gothenburg area. We will mention this in the MM section (see response to comment below). We believe that it is very hard to say something quantitatively about the possible influence of the summer temperatures on the Hg levels, but it is a good thing to mention that this summer was not typical with respect to temperature. F) The important thing is that the September values represent the concentration of Hg when the leaves are (soon to be) shed, which is the flux from foliage to soil. G) We do not agree that our C+3 needles represent four years of uptake. Needles were sampled by the end of June as explained in the manuscript. In evergreen conifers the new flush of needles start develop in the end of spring and become fully developed in early summer. Thus, our C+1 needles were just slightly more than one year when they were sampled and the C+3 just over three years. If we would have sampled the C+3 needles by the end of October they would have seen four growing seasons, but, as explained, this was not the case.*

Line 21: C + 3 represents three-year old needles.

**Response:** *What we refer to here is the literature data (Figure 4), which covered up to C+4 needles. However, we have not expressed ourselves very clearly here and will add "in the literature data" to this sentence.*

Line 23: approximately how much older is foliage from Rwanda compared to foliage from Gothenburg in this study?

**Response:** *On line 305-306 we say: "The seasonal accumulation of Hg was not possible to determine for the Rwandan trees, as the leaf ages were unknown, but most of them were likely in the range of 0.5 to 2 years." Thus, we believe that we have answered this in the main text of the manuscript. We think it is unnecessary to go into further detail in the Abstract, which should be a condensed piece of text.*

Line 25 – 29: please be careful about possible correlations of [Hg]$_A$ with SLA (see general comment above)

**Response:** *We still believe that our data suggest such a relationship, but that our phrasing of the last part of the Abstract (and other places) can be modified not to overstate its generality. Therefore, we now write: "To search for general patterns, the accumulation rates of Hg in the diverse set of tree species in the Gothenburg area were related to the specific leaf area (SLA). Leaf area based [Hg] was negatively and non-linearly correlated with SLA, while mass-based [Hg] had a somewhat weaker positive relationship with SLA. An elaborated understanding of the relationship behind [Hg] and SLA may have the potential to support large-scale modelling of Hg uptake by vegetation and Hg circulation."*

Line 41 – 44: measuring litterfall Hg is time-intensive but not particularly challenging. Rather just describe all Hg dry deposition pathways.

**Response:** *Yes, we removed this part of the sentence: ", however, is more challenging and".*

Lines 63 – 64: please be more specific, which processes you mean. Do you refer to biochemistry inside the leaf or to litterfall deposition? Agnan et al. 2016 might not be the most fitting publication here.

**Response:** *OK, thanks, maybe not completely clear. We have changed the wording: "The detailed character and relative importance of biochemical processes inside the leaf subsequent to stomatal uptake in the accumulation of Hg in leaf tissue are not completely understood. Du & Fang (1983) found that in wheat leaves $Hg^0$ was converted to divalent $Hg^{2+}$ resulting from oxidation, likely promoted by the enzyme catalase." The Agnan et al reference has been removed.*

Lines 66 – 71: this paragraph is a bit lengthy. It would be enough to cite abundant literature, that root Hg uptake is small compared to foliar Hg uptake from the air.

**Response:** *This piece of text has been changed based on a comment by Anonymous Reviewer 1 with one reference replaced by another etc. We have also removed some unnecessary text.*

Line 81: please reason, why $[Hg]_A$ is more relevant than $[Hg]_M$.

**Response:** *We do not say that $[Hg]_A$ in general is more relevant than $[Hg]_M$, but that in some analyses on ecosystem scale it can be more relevant. We believe that we already answer the question why $[Hg]_A$ can be more important is the following piece of text: "The significance of $[Hg]_A$ follows from the assumption that $Hg^0$ is mainly taken up through the leaf surface i.e. stomata and cuticles as explained above. The ecosystem uptake may therefore partly be driven by its leaf area index (LAI, unit area leaves per unit area ground). However, if [Hg] in the leaves saturates with time, the ecosystem uptake might instead be limited by the total leaf mass." (line 83-86)*

Line 84: please add citation

Line 91: please add citation

**Response:** *We believe that the statements on line 84 and line 91 discuss the potential importance of different processes based on premises presented elsewhere in the text. As such, they do not need references. For example, the argument that if leaf mass is limiting, species with large leaf mass has the capacity to take up more Hg is quite basic.*

Line 90 – 92: please be more clear, that you are referring to deciduous and coniferous forests with the same LAI

**Response:** *Agree. We have changed the wording: "… than broadleaved, provided that the two forests have the same LAI, thus possibly representing a stronger Hg sink."*

Line 94 – 96: please elaborate on this in more detail, since it might not be clear for every reader, why SLA is needed for biogeochemical cycling.

**Response:** *In plant ecophysiology it is customary to search for relationships between processes (such as Hg accumulation) and leaf traits. SLA is a key leaf trait. To clarify, we have modified the sentence: "If the unit leaf area accumulation of Hg depends on the fundamental leaf characteristic SLA, a general understanding of the relationship behind [Hg]$_A$ and SLA would support improved descriptions of Hg uptake by vegetation applied in large-scale atmospheric modelling and assessments of its biogeochemical cycling." In addition, the significance of SLA has already been explained earlier in the Introduction.*

Line 116 – 119: the Gothenburg area has a cool maritime climate, however, was this true for the unusually hot and dry European summer 2018? It would be beneficial to give actual weather data of this particular year here.

**Response:** *Yes, the Gothenburg area experienced higher temperature than normal in 2018. We have added the following piece of text: "Like large areas of north and central Europe, Gothenburg experienced an unusually warm and dry summer in 2018. The average daytime temperature from April–September 2018 was higher by 2.0–2.4$^o$C in 2018 and the water vapour pressure deficit (VPD) was positively affected compared to the preceding five-year period (Johansson et al., 2020)."*

Line 125: consider moving Table 1 to SI

**Response:** *When we drafted the manuscript the information presented in this table was embedded in the text. However, a better overview was obtained combining the information in a table. We considered to have the table as supplementary, but we believe that the information about species included etc in the main text would then be too limited. Thus we wish to retain Table 1 in the main text.*

Line 132 – 139: you could shorten/move to SI the description of Nyungwe forest and instead give some details about tree physiology of interest at this forest (e.g. how old is foliage there on average?)

**Response:** *We have shortened the text about Nyungwe forest to: "Nyungwe tropical montane rainforest is located in South-Western Rwanda covering 1013 km$^2$ (Table 1). It consists of a mixture of late successional and early successional trees, mainly evergreens.*

*Annual mean temperature in sampled areas is 13.7 to 15.6°C and mean annual precipitation 1867 mm. The seasonal variation in temperature is small but precipitation varies spatially and seasonally, with a dry period of two months, from mid-June to mid-August. Sampled trees were growing at elevations between ca 1950 and 2500 m a.s.l. along a 32 km-long east-to-west transect. Nyirambangutse et al. (2017) provides details of the sampling sites and characteristics of the tree species."*

Line 159: I think you can move your results on the effect of drying temperature on Hg concentrations from the Discussion (Line 452 – 456) to Sect. 2.2, to make these results more accessible to the readership. Consider moving Figure 1 to SI to shorten the paper.

**Response:** *Agree, we have moved the suggested text from the Discussion to Section 2.2. We have retained the figure in the main text. The space saved by moving it to supplementary is small.*

Line 200 – 203: I challenge the representativeness of $[Hg]_A$ September values for the whole growing season (see general comment above)

**Response:** *The September values should be representative of the $[Hg]_A$ of the leaves when they are shed to the ground in the autumn, which represents the flux of Hg from foliage to soil (maybe a slight underestimation because of some additional uptake during the short period from sampling to senescence and shedding).*

Line 220 – 221: please exclude Cryptomeria $[Hg]_A$ from the study. Significant differences of $[Hg]_A$ between Cryptomeria and Abies are unsurprising given that surface area measurements of Cryptomeria cannot be compared to the other species.

**Response:** *As explained above, Cryptomeria data has been removed except for Figure 2a, which does not depend on SLA.*

Line 242 – 244: however, slope in Figure 4b is < 1, thus there seems to be a slight decrease of Hg uptake with needle age

**Response:** *We agree in principle, but since C+1 (the second needle age class) is used as the reference needle age class, theoretically the slope coefficient would be 0.5. Now it is 0.436 and thus the comment is still valid to a certain extent. Consequently, we add the following sentence at the end of the paragraph "However, since the slope coefficient of the relationship presented in Figure 4b is <0.5 there is still an indication of a decrease in the rate of Hg accumulation over time."*

Line 255 – 256: what exactly is significantly different between the mentioned species? The slope of [Hg] with time (June to September)?

**Response:** *Rather than the slope (we are using ANOVA here, not regression) it is the difference in $[Hg]_M$ and $[Hg]_A$, respectively, between the mentioned species which is significant. This is the information that can be obtained from the mixed design ANOVA and the post-hoc test. To make this absolutely clear we have changed the start of the relevant sentence to: "The post-hoc test of the ANOVA showed that …".*

Line 260: Typo (Agee, Fig. 5a)

**Response:** *Thanks, corrected.*

Line 267: please be more specific. What is this genetic variation and why does it affect foliar Hg uptake?

**Response:** *To exemplify the type of inter-species variation that is likely to be of importance we have changed the last sentence of this paragraph to: "The genetic variation in properties of interest for Hg uptake, such as inter-species variation in stomatal conductance, is expected to be small when only one species is considered."*

L278: C+3 needles should be divided by the factor of 4 to be comparable with broad leaves, as they were accumulating Hg over 4 consecutive growing seasons.

**Response:** *As stated above, we do not agree that our C+3 needles represent four years of uptake. Needles were sampled by the end of June as clearly explained in the manuscript. In evergreen conifers the new flush of needles start develop by the end of spring and become fully developed in early summer. Thus, our C+1 needles were just slightly more than one year when they were sampled and the C+3 just over three years.*

Line 307: please be more precise about statistical significance. Which regression model did you use and which regression parameter was significant?

**Response:** *This information has been deleted, see next point.*

Line 308: was there a significant regression coefficient of [Hg] to SLA for broad leaf values from Gothenburg? You indicate that there should be a correlation, but this is not clear to me.

**Response:** *We agree that the phrasing may be misleading and have removed the sentence: "No significant relationship between $[Hg]_M$ and SLA was observed ($R^2 = 0.03$, $p = 0.45$), possibly reflecting the much more narrow range of SLA for the Rwandan trees compared to those in the Gothenburg area".*

Line 328: please elaborate on this further. By how much time was the period of gas exchange shortened?

**Response:** *In this particular experiment gas exchange was not measured, but the duration of the grain filling and the longevity of flag leaves were investigated based on observations performed three days per week (flag leaf chlorophyll concentration and of growth of grains). These response variables also highlighted the senescence-promoting effect of ozone. The effect of ozone exposure on gas exchange has been investigated extensively and this sentence refers to this general understanding. To become more explicit we have added references and one sentence related to the observation on the senescence effect by ozone in the specific experiment: "The magnitude of this leaf age effect on $[Hg]_M$ declined with increasing ozone concentration, likely associated with the shortening of leaf life span from ozone (Mullholland et al., 1998), reducing the duration of the period with gas exchange of*

the leaves (Mullholland et al., 1997; Feng et al., 2010; Osborne et al. 2019). Gelang et al. (2000) found, for the experiment used here, a reduction in the duration of grain filling and a reduced flag leaf (the uppermost leaf on the shoot) area duration, based on chlorophyll measurements, which were strongly related to ozone exposure. Compared to the NF treatment, the duration of the flag leaf was estimated to be reduced by 5, 19 and 29 days in the NF+, NF++ and NF+++ treatments, respectively." We do not believe that this manuscript is the place to go deeper into details of the physiological responses of wheat to ozone.*

Line 336: in case length is a problem, please shorten this section and avoid redundancy.

**Response:** *We have removed the second half of the first paragraph of the Discussion.*

Line 352 – 355: as an additional factor litterfall could have lost organic carbon as well, therefore concentrating Hg (see e.g. Pokharel and Obrist, 2011)

**Response:** *Yes, this process may be important. We added this sentence to the relevant paragraph: "It should then be kept in mind that litterfall could have lost organic carbon, therefore concentrating Hg (e.g. Pokharel and Obrist, 2011) and that the variation in data compiled by Wang et al. (2016) was substantial." (The link to Wang et al depends on a response to Anonymous Reviewer 1. There text has also been structured differently here because of comments from Anonymous Reviewer 1.)*

Line 359 – 361: I disagree, needle litterfall flux is determined as Hg per unit ground area and it does not matter how old litterfall needles are, thus there is no overestimation. Which values do you compare to needle litterfall flux here, please elaborate.

**Response:** *It is predominantly the oldest needles that are shed, but maybe we did not express ourselves sufficiently clearly here. What we mean is that if Hg concentrations of needles are used to estimate litter flux of Hg there may be a bias if data only for young needles are used. If litterfall is measured directly this bias is not a problem. We have changed the sentence and added a further sentence: "We conclude that the bias towards sampling younger leaves/needles that had less time to accumulate Hg could, if used to estimate litter Hg flux, leads to an underestimation of the flux of Hg by leaf/needle shedding. Direct measurements of litter flux do not suffer from this bias."*

Line 385 - 386: I do not fully agree with this statement, fluxes are normalized over time, cancelling out high [Hg] in older foliage when we are preferentially interested in fluxes over the course of one growing season. Please also consider that the mass of younger foliage at trees is usually higher compared to older foliage. Therefore, uptake into younger foliage is more important for the overall flux and for this reason younger foliage should arguably be monitored preferentially when we aim to assess fluxes.

**Response:** *We are not sure that we understand this comment completely. On line 385-386, we are discussing tropical trees with long-lived leaves. We believe that a better understanding of the development of $[Hg]_M$ with leaf age would help quantifying Hg accumulation in tropical trees. In the autumn of 2020 we sampled three leaf age classes of two broadleaved trees with perennial leaves in the Botanical Garden of Gothenburg and*

*found a more or less linear increase in [Hg]$_M$ with leaf age (data to be published later). Thus, the accumulation of the full leaf life span of evergreen broadleaved trees should be taken into consideration.*

Line 397 – 398: I challenge this statement of SLA being a strong predictor from the regression presented in Fig. 7 (see general comment above)

**Response:** *We agree that our statement may be too strong. We have changed the sentence to: "However, it means that SLA as a leaf trait might be a useful predictor of Hg accumulation, which should be investigated further, and points towards the significance of not neglecting …"*

Line 404: …the same projected leaf area over unit ground area

**Response:** *Thanks, changed as suggested.*

Line 407: Pleas add a reference here to back up the statement. When does gas exchange of needles typically end in the Gothenburg area? Please keep in mind that bud break is typically later of needles than leaves.

**Response:** *The gas exchange of conifers starts already in March and ends in November in the Gothenburg region, but due to the relatively low temperature in March and the short daily photoperiod in November the total net assimilation in these month is low but still significant (Tarvainen et al 2015). Yes, bud break in needles is somewhat later than leaf emergence, but this applies only to current year (C) needles. Conifers often retain their needles 4-10 years, depending on species and location, and for leaves older than C the duration of the season with gas exchange is longer than for deciduous trees.*

Line 412: please include stomatal conductance values

**Response:** *We believe that the reference provided is sufficient. To start specifying numerical values of stomatal conductance ranges for different plant types should not be necessary. Anonymous Reviewer 1 has asked us to shorten the Discussion and to provide this type of details will instead extend the Discussion.*

Line 433 – 465: please shorten this part and avoid repetition

**Response:** *We have cut the text in this part by approximately 50%.*

Line 464: please be more precise, which had been assumed earlier?

**Response:** *This text has now been deleted because of the reduction of the text mentioned in the previous point.*

Line 469: 3 years old

**Response:** *Changed to three years old (although the literature data contained information of needles up to four years old).*

Line 472 – 473: I disagree, see comment for line 385 – 386

**Response:** *We believe that what we say here is essentially correct. To become more precise we have however changed the phrasing: "A bias towards sampling younger needle age classes of conifers, if used to assess Hg flux from foliage to soil, can lead to underestimation of this Hg flux since it is predominantly the oldest needles that are shed as litter."*

Line 476 – 480: I challenge this (see general comment above)

**Response:** *We still believe that our data suggest that such a relationship exists. However, we do not wish to overstate our observation and have moderated this conclusion point: "Specific leaf area (SLA) is a leaf trait, which showed a relationship with unit area accumulation of Hg over the wide range of leaves/needle characteristics covered by our study of trees in the Gothenburg area. There was a weaker positive relationship between leaf/needle mass based uptake of Hg and SLA. Studies should be undertaken to further investigate the role of SLA in Hg accumulation of leaves and needles."*

*References:*

*Poorter et al. 2009: Causes and consequences of variation in leaf mass per area (LMA): a meta-analysis. New Phytologist*

*Tarvainen et al 2015: Seasonal and within-canopy variation in shoot-scale resource use efficiency trade-offs in a Norway spruce stand. Plant Cell and Environment 38, 2487-2496.*

*Wohlgemuth et al. 2020: A bottom-up quantification of foliar mercury uptake fluxes across Europe. Biogeosciences*

---

## Referee Report (RR1)

The authors put careful work into the revision of the manuscript, which improves the scope of the paper. I therefore support the publication of the manuscript as it is.

Two final remarks for clarification:

- Line 385 – 386, first submission: I agree, that we have to be mindful about foliage age when we evaluate e.g. foliar Hg concentrations. Just for the net Hg(0) flux (e.g. to one forest site over one growing season) foliage age might not be as relevant because the foliage Hg uptake rates normalized over time are not so different between old and young foliage (Fig. 4b first submission) and the mass of younger foliage is often much higher than the mass of older foliage at the same tree (compare e.g. masses of spruce needles current season to six year old needles). I just think we have to carefully differentiate in wording between concentrations and fluxes so they do not get confused.
- Even though SLA values of coniferous needles and broad leaves are usually quite distinct, I agree that there is a large SLA variation within these categories and sometimes we observe inbetween SLA values (e.g. by Larix). My suggestion to separate needles and leaves is more born out of their different metabolic strategies and structure, which might impact Hg uptake. I am excited for further insights into the similarities between uptake mechanisms of needles and leaves. Whatever the direction, SLA seems to be a functional trait of interest in this context.

---

## Author Response (AR2)

**Response to reviewers – second review**

**Comments by reviewer Lena Wohlgemuth**

The authors put careful work into the revision of the manuscript, which improves the scope of the paper. I therefore support the publication of the manuscript as it is. Two final remarks for clarification:

• Line 385 – 386, first submission: I agree, that we have to be mindful about foliage age when we evaluate e.g. foliar Hg concentrations. Just for the net Hg(0) flux (e.g. to one forest site over one growing season) foliage age might not be as relevant because the foliage Hg uptake rates normalized over time are not so different between old and young foliage (Fig. 4b first submission) and the mass of younger foliage is often much higher than the mass of older foliage at the same tree (compare e.g. masses of spruce needles current season to six year old needles). I just think we have to carefully differentiate in wording between concentrations and fluxes so they do not get confused.

**Response:** W*e agree that one has to differentiate between concentrations and fluxes, but concentrations are relevant to fluxes.*

• Even though SLA values of coniferous needles and broad leaves are usually quite distinct, I agree that there is a large SLA variation within these categories and sometimes we observe inbetween SLA values (e.g. by Larix). My suggestion to separate needles and leaves is more born out of their different metabolic strategies and structure, which might impact Hg uptake. I am excited for further insights into the similarities between uptake mechanisms of needles and leaves. Whatever the direction, SLA seems to be a functional trait of interest in this context.

**Response:** *We completely agree on this.*

**Comments from the anonymous reviewer**

Comments on "Mercury accumulation in leaves of different plant types – the significance of tissue age and specific leaf area" (BG-2021-117).

Pleijel et al. studied the Hg concentrations in the leaves of deciduous, conifer and evergreen trees and wheat and found that the accumulation rates of Hg in the diverse set of tree species were related to the specific leaf area. I think this is interesting. Additionally, leaf area based [Hg]A is relevant to assess the tree canopy accumulation capacity of Hg, typically being substantially higher in conifers than in deciduous trees. I recommend minor revise before acceptance.

Line 35. I donot think the background concentrations of GEM in Northern hemisphere is approximately 1.5 - 1.7 ng m-3. I think the concentrations are much lower. Please double –check and the Sprovieri et al. (2016) seems did not show the background concentrations of GEM in Northern and southern hemisphere.

**Response:** *We have revisited the paper by Sprovieri et al (2016), who state that "The GEM concentrations highlight that the mean GEM values of most of the GMOS sites were between 1.3 and 1.6 ng m$^{-3}$, with a typical interquartile range of about 0.25 ng m$^{-3}$." We have accordingly adjusted the concentration range to 1.3 and 1.6 ng m$^{-3}$. We do not believe that it is correct to say that the GEM background concentrations in the Northern Hemisphere is "much lower" than 1.5 - 1.7 ng m$^{-3}$.*

Line 55. New article by Zhou et al (2021) (Vegetation uptake of mercury and impacts on global cycling) has reviewed the Hg source in vegetation, including leaf and needle, which would help you to know the leaf Hg uptake and emission.

**Response:** *We agree that this reference is relevant. Unfortunately, we did not observe this article since it was published just before we submitted our manuscript. Thank you for drawing our attention to this valuable article.*

*We have now referred to this paper in an already existing sentence in the Introduction. In addition, the second sentence of Discussion now reads "This was further substantiated by the review of Zhou et al. (2021)"., and in the support to our observation of higher $[Hg]_M$ in the tropical trees compared to the Swedish deciduous trees we added: "… and the findings by Zhou et al. (2021) that evergreen broadleaf trees have significantly higher $[Hg]_M$ compared to deciduous trees." (see line 399-400 in revised version).*

Line 105-110. Based on my knowledge, all the four hypotheses have been confirmed by previous studies. What's the new perspectives of current study?

**Response:** *We believe that the relationship between Hg accumulation and SLA has been investigated only to a very limited extent earlier (Hypothesis 2). To do this it was necessary to obtain data on Hg accumulation over time (related to Hypothesis 1) for a wide range of species with different SLA. The amount of data on leaf concentrations of Hg on tropical evergreen species (especially from Africa) is small and the comparison with temperate species is relevant to the field (Hypothesis 3). In relation to Hypothesis 4, we believe that the observation of the pronounced effect of charcoal filtration of air lead to a strongly reduced leaf Hg concentration of field-grown wheat. Also, the effect of ozone on leaf Hg concentration by reducing the leaf life span is new and relevant to the field.*

Line 380. I donot think the higher Hg in litterfall is due to organic matter loss. As the author observed higher Hg concentrations in older foliage. Litterfall lives longer time than foliage, which is the main reason for the high Hg concentration.

**Response:** *This is not the topic on line 380, neither in the clean manuscript, nor in the annotated version. We assume that the comment refers to the text on lines 383-384 in the annotated version. In response to the comment we have added the following sentence: "Also, it cannot be excluded that Hg accumulation in litter continues after shedding of leaves/needles."*

Line 515. Besides life span, what the other reasons for the foliage Hg differences?

**Response:** *"Life span" is not mentioned on Line 515, neither in the clean version of the manuscript, nor in the annotated. We assume that the comment refers to the text on line 525 in the annotated version of the manuscript. In response to the comment by the reviewer we have added this piece of text here: ", but differences in rate of gas exchange may also be of importance."*